# Normal cells repel WWOX-negative or -dysfunctional cancer cells via WWOX cell surface epitope 286-299

Yu-An Chen[1,5], Yong-Da Sie[1,5], Tsung-Yun Liu[1,5], Hsiang-Ling Kuo[1,5], Pei-Yi Chou[1], Yu-Jie Chen[1], Kuan-Ting Lee[1], Pin-Jun Chen[1], Shur-Tzu Chen[2] & Nan-Shan Chang [1,3,4 ✉]

Metastatic cancer cells are frequently deficient in WWOX protein or express dysfunctional WWOX (designated WWOXd). Here, we determined that functional WWOX-expressing (WWOXf) cells migrate collectively and expel the individually migrating WWOXd cells. For return, WWOXd cells induces apoptosis of WWOXf cells from a remote distance. Survival of WWOXd from the cell-to-cell encounter is due to activation of the survival IκBα/ERK/ WWOX signaling. Mechanistically, cell surface epitope WWOX286-299 (repl) in WWOXf repels the invading WWOXd to undergo retrograde migration. However, when epitope WWOX7-21 (gre) is exposed, WWOXf greets WWOXd to migrate forward for merge. WWOX binds membrane type II TGFβ receptor (TβRII), and TβRII IgG-pretreated WWOXf greet WWOXd to migrate forward and merge with each other. In contrast, TβRII IgG-pretreated WWOXd loses recognition by WWOXf, and WWOXf mediates apoptosis of WWOXd. The observatons suggest that normal cells can be activated to attack metastatic cancer cells. WWOXd cells are less efficient in generating $Ca^{2+}$ influx and undergo non-apoptotic explosion in response to UV irradiation in room temperature. WWOXf cells exhibit bubbling cell death and $Ca^{2+}$ influx effectively caused by UV or apoptotic stress. Together, membrane WWOX/TβRII complex is needed for cell-to-cell recognition, maintaining the efficacy of $Ca^{2+}$ influx, and control of cell invasiveness.

[1] Institute of Molecular Medicine, National Cheng Kung University, Tainan, Taiwan, Republic of China. [2] Department of Cell Biology and Anatomy, National Cheng Kung University, Tainan, Taiwan, Republic of China. [3] Advanced Optoelectronic Technology Center, National Cheng Kung University, Tainan, Taiwan, Republic of China. [4] Graduate Institute of Biomedical Sciences, College of Medicine, China Medical University, Taichung, Taiwan, Republic of China. [5] These authors contributed equally: Yu-An Chen, Yong-Da Sie, Tsung-Yun Liu, Hsiang-Ling Kuo. ✉email: wox1world@gmail.com

Human WWOX gene encodes a 46-kDa tumor suppressor protein WW domain-containing oxidoreductase, designated WWOX, FOR, or WOX1[1–6]. WWOX participates in the regulation of many signal pathways (e.g., p53, TGF-β, Wnt, Hippo, YAP/TAZ, Hyal-2, etc.) for cell growth and differentiation and organ development[7–13]. Alterations in these pathways due to WWOX dysfunction may facilitate disease progression[7–13]. Aberrant WWOX phosphorylation also enhances the progression of cancer and Alzheimer's disease (AD)[9,14–16]. For example, transition of proapoptotic pY33-WWOX to prosurvival pS14-WWOX promotes the progression of both cancer[15,16] and AD[17]. Human newborns deficient in WWOX suffer severe neural diseases, metabolic disorders, and early death, but fail to form tumors spontaneously[14,18–23]. When functional WWOX starts to downregulate in the brain of middle-aged healthy individuals, this may lead to slow aggregation of a cascade of brain proteins that eventually form tau tangles and amyloid beta plaques in old ages[14,24–28]. WWOX has recently been established as one of the AD risk factors[29]. As a partner of WWOX binding proteins, p53 may functionally counteract with WWOX that leads to neural inflammation and aggregate formation of tau, amyloid beta and other proteins in vivo as shown in the AD pathologies[30]. WWOX gene is located on a common fragile site FRA16D on chromosome ch16q23.3-24.1, alterations of this gene is associated with cancer development[1–6]. Loss of heterozygosity (LOH) of human WWOX gene may go as high as 30–50%. Human WWOX gene is rarely mutated. Majority of metastatic cancer cells are deficient in WWOX protein due to epigenetic modification[1–6] and translational blockade of WWOX/Wwox mRNA[31].

Tumor suppressor WWOX is anchored, in part, in the cell membrane by Hyal-2, as determined by immunoelectron microscopy[7,13,32]. Also, membrane WWOX may undergo self-polymerization[7,32]. Proapoptotic pY33-WWOX binds Hyal-2 and Smad4[7,32]. When the signaling complex WWOX/Hyal-2/Smad4 is transiently overexpressed, TGF-β or hyaluronan induces internalization of the signaling complex to relocate the nucleus for causing cell death[7,32]. Most recently, we have identified two cell surface-exposed WWOX epitopes, namely WWOX7-21 and WWOX286-299[16,33]. Synthetic WWOX7-21 peptide potently enhances ceritinib-mediated breast 4T1 cell death[33]. Also, WWOX7-21 peptide significantly enhances ceritinib-mediated explosion and death of 4T1 stem cell spheres[33]. The proapoptotic effect of WWOX7-21 peptide is due to (1) reduction of ERK phosphorylation, (2) significant upregulation of pY33-WWOX, (3) rapid increases in Ca²⁺ influx, and (4) disruption of the IκBα/WWOX/ERK prosurvival signaling[33]. In stark contrast, pS14-WWOX7-21 peptide strongly supports cancer growth in vivo and blocks ceritinib-mediated apoptosis in vitro[33]. Overall, endogenous pY33-WWOX is proapoptotic and acts as a tumor suppressor[4,31,34–36]. In contrast, pS14-WWOX strongly supports lymphocytic cell differentiation[9,37] but enhances the progression of cancer[15,16] and AD[17].

A majority of metastatic cancer cells are devoid of WWOX expression or possess dysfunctional WWOX protein[1–6]. These cells tend to have an enhanced mobility and increased sensitivity to microenvironment[38]. For example, when WWOX-deficient or -dysfunctional cells (hereby designated WWOXd) encounter functional WWOX-expressing (WWOXf) cells, WWOXd cells migrate forward and then backward to avoid physical contacts with WWOXf cells[38]. WWOXf cells are mainly normal cells, and WWOXd are metastatic cancer cells. To survive in the WWOXf microenvironment, WWOXd cells strongly induce the redox activity in WWOXf cells to cause apoptosis, and then secrete transforming growth factor beta (TGF-β) to compromise with the surviving WWOXf cells in target organs for docking and homing[38]. Here, we examined whether cell surface WWOX epitopes are involved in the cell-to-cell recognition. Real-time cell migration analysis revealed that WWOX7-21 (gre) is responsible for cell-to-cell greeting. WWOX286-299 (repl) is able to repel WWOXd cells. Further, we determined here that in response to stress stimuli at room temperature, WWOXf cells undergo non-apoptotic bubbling cell death (BCD) initiating from the nucleus at room temperature[39,40]. WWOXf cells have a functional Ca²⁺ influx system. In contrast, WWOXd cells have a less efficient Ca²⁺ influx system and the cells undergo pop-out explosion death by stress stimuli at room temperature. Overall, our study provides evidence that WWOXf cells such as normal cells have an intact system for Ca²⁺ influx, tend to undergo stress-regulated BCD, and repel visiting WWOXd cells via cell the surface repl epitope.

## Results

**WWOXf cells repel WWOXd cells during encounter.** Upon sensing WWOXf cells from a distance of 500 μm or less, WWOXd cells accelerate their migration toward the WWOXf cells. However, both cells try to avoid physical contacts[38]. WWOXd cells increase the redox activity of WWOXf cells to cause apoptosis from a distance[38]. Where indicated, a selected pair of cells were grown, respectively, in the left and right chambers of a culture insert (ibidi) using 2% or 10% FBS/medium for 24–48 hr (Supplementary Fig. 1a in the section A of Supplementary Figures of Supplementary Information). 2% FBS/medium was to support cell survival and migration, but did not support cell division (Supplementary Fig. 1b)[38]. When wild type and Wwox knockout MEF cells were in the coculture, the MEF Wwox knockout cells had an accelerated migration speed and overall migration distance and underwent retrograde migration (Supplementary Fig. 1c–f). In most cases, WWOXd cells divide when they migrate relatively close to WWOXf cells (~10–100 μm). One of the divided WWOXd daughter cells moved forward with an elongated protrusion to probe the WWOXf cells, whereas the other daughter cell moved back to the original home base[38]. When cells were cultured in 10% FBS/medium for migration assay, the Wwox knockout MEF cells underwent retrograde migration upon facing wild type MEF cells (Supplementary Fig. 1g and Supplementary Video 1). Many wild type cells underwent apoptosis (Supplementary Video 1). The legends for all videos are at the Description of Additional Supplementary Information.

In addition, when WWOXf TNF-sensitive L929S fibroblasts[3,7] and WWOXd TNF-resistant L929R cells[3,7] were paired up for migration assay under 10% FBS, retrograde migration of L929R cells occurred and the mobility of L929S cells was dramatically reduced (Fig. 1a–c; Supplementary Video 2). When the serum concentration was decreased down to 0% and 2%, increased mobility of L929S cells was observed (Fig. 1e–g, i–k), suggesting that certain serum factors limit L929S migration.

**WWOXf cells are susceptible to WWOXd cell-mediated apoptosis and UV/cold shock-induced Ca²⁺ influx and bubbling cell death (BCD), whereas WWOXd cells undergo explosion by UV/cold shock.** WWOXd cells induce apoptosis of WWOXf cells from a distance of <500 μm[38]. For example, when L929R faced L929S during migration, L929R cells caused apoptosis of L929S cells without physical contacts. Compared to L929R cells, apoptosis of L929S cells was about 1.5-fold increases (Fig. 1d, h, l). Also, migrating MDA-MB-435s cancer cells induced apoptosis of MEF wild type cells (Supplementary Video 3). Many MEF wild type cells underwent typical apoptosis, including membrane blebbing, whole cell condensation and death. Cell death also occurred in few MDA-MB-435s cells.

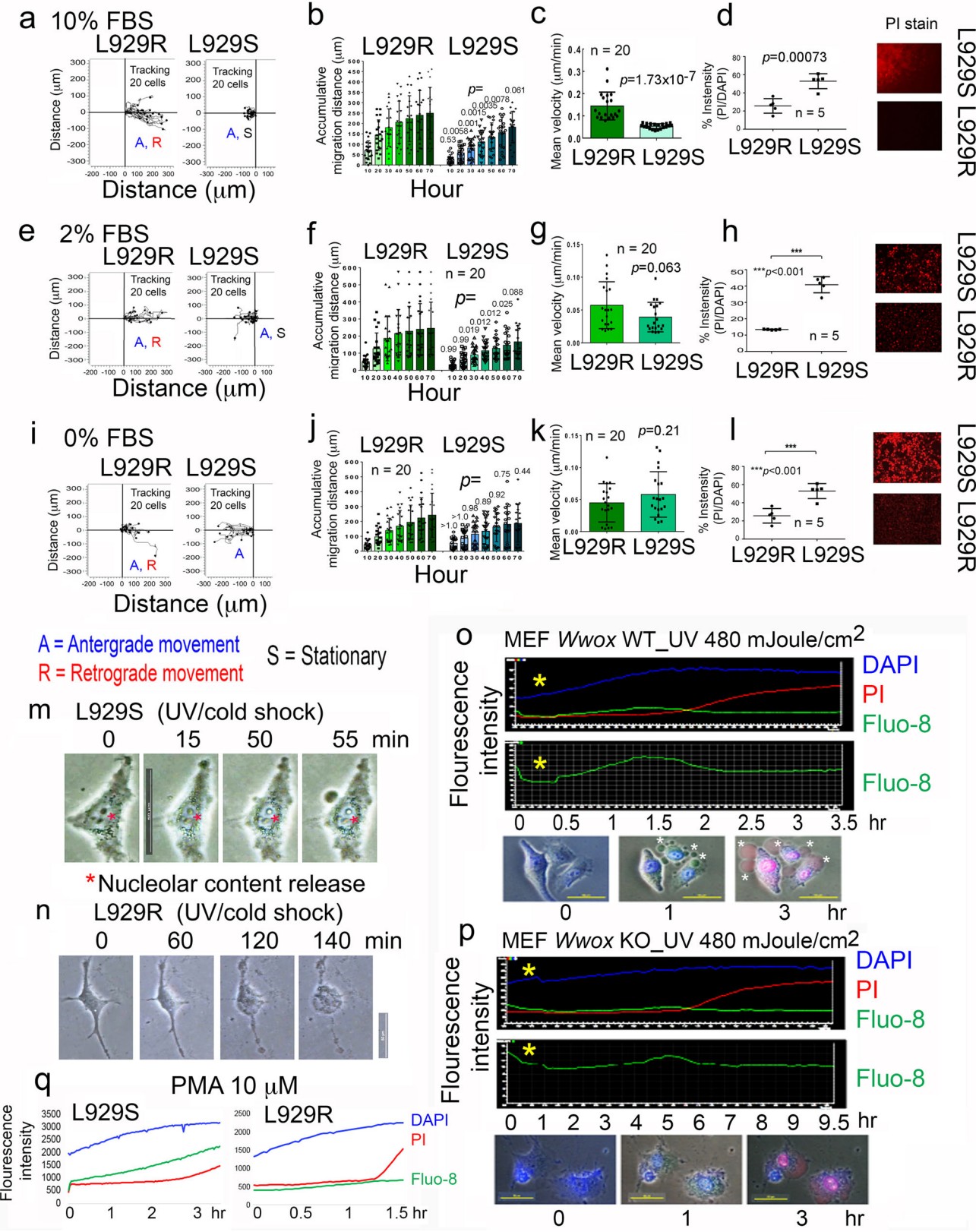

A = Antergrade movement
R = Retrograde movement
S = Stationary

*Nucleolar content release

Next, we examined the functional differences between WWOXf and WWOXd cells. We reported that when WWOXf cells are exposed to UV irradiation, or UV and subsequent cold shock at 4 ℃ for 5 min, these cells undergo non-apoptotic BCD at room temperature[39]. Cold shock is intended to enhance the effect of UV[39]. During BCD, each cell generates one or two nitric oxide-containing nuclear bubbles (or balloons possessing short stalks) for release or hanging onto the cell surface. No caspase activation was observed[39,40]. When L929S cells were exposed to UV (480 mJoule/cm$^2$) followed by exposure to cold shock at 4 ℃ for 5 min, the cells underwent bubbling first, then Ca$^{2+}$ influx and finally cell death (Fig. 1m; Supplementary Fig. 2a and

**Fig. 1 TNF-sensitive L929S cells induce TNF-resistant L929R cells to undergo retrograde migration, whereas L929R cells cause L929S to undergo apoptosis.** Both L929S and L929R cells were cultured in 10% FBS/RPMI medium overnight in the right and left chambers of a culture insert (from ibidi), respectively. **a–d** During time-lapse microscopy for cell migration, cells were cultured in 10% FBS/RPMI containing nontoxic propidium iodide (PI; 7.5 μM/ ml). The accumulative migration distance and mean velocity were calculated (**b**, **c**). The extent of cell death was calculated based upon PI uptake (**d**). One-way Anova was used for statistical analysis. **e–l** Cell migration assays were carried out in 2% and 0% FBS, respectively. **m** When L929S cells were exposed to UV (960 mJoule/cm$^2$) and subsequent cold shock at 4 °C for 5 min, the cells underwent BCD. Also, see Supplementary Video 4. Release of nucleolar content is shown. **n** Under similar treatment, L929R cells underwent explosion. Also, see Supplementary Video 5. **o**, **p** In response to UV, MEF wild type cells underwent BCD, and MEF *Wwox* knockout cells exploded. **q** PMA induced Ca$^{2+}$ influx in L929S cells, but not in L929R cells.

Supplementary Video 4). Fluo-8 was used to measure the calcium influx in live cells.

In response to UV, L929R cells underwent pop-up explosion within approximately in 1 hr, followed by uptake of propidium iodide (PI), indicating the occurrence of cell death. Low efficiency of Ca$^{2+}$ influx was observed (Fig. 1n; Supplementary Fig. 2a; Supplementary Video 5). Normal skin fibroblasts from a patient YMY underwent Ca$^{2+}$ influx and then BCD upon exposure to UV (Supplementary Fig. 2b). In contrast, human neurofibromatosis YMY-NF1 cells exploded upon treatment with UV (Supplementary Fig. 2b). When cells were exposed to UV, these cells rapidly picked up DAPI in the cytoplasm. DAPI was rapidly accumulated in the nucleus due to UV-increased nuclear membrane permeability.

In response to UV irradiation, MEF Wwox wild type cells underwent Ca$^{2+}$ influx and BCD effectively (Fig. 1o). Under similar conditions, MEF *Wwox* knockout cells underwent explosion, and calcium influx in these cells were less effective (Fig. 1p). By using phorbol myristate acetate (PMA) to mobilize calcium[41], dramatic Ca$^{2+}$ influx was shown in L929S cells, but not in L929R cells (Fig. 1q).

**Screening a panel of WWOXf and WWOXd cells for BCD or explosion**. Our data showed that WWOXf cells were responsive to induction of Ca$^{2+}$ influx and BCD by stress stimuli such as UV, ceritinib[33], sodium orthovanadate (Na3VO4), hydrogen peroxide, proteinase inhibitor cocktail, and estrogen receptor agonist CI-4AS-1 (Supplementary Fig. 2a–c). The following WWOXf cell lines were used, including human breast MCF7 cancer cells[35], human colon HCT116 cancer cells, human prostate cancer DU145 cells, murine L929S fibroblasts[3], mink lung Mv1Lu epithelial cells, testicular cancer NT2D1 cells, human neuroblastoma SH-SY5Y cells, human non-small cell lung NCI-H1299 carcinoma cells, human tongue squamous cell carcinoma SCC4, SCC15, and SCC9[42], and normal human skin fibroblasts (Supplementary Fig. 2a–c).

In contrast, WWOXd cells were less responsive to the induction of Ca$^{2+}$ influx and underwent explosion in response to the aforementioned stress stimuli (Supplementary Fig. 2a, b, d). These WWOXd cells included human breast MDA-MB-231 cancer cells[35], human MDA-MB-435s cancer cells[35], human neuroblastoma NB69 cells, mouse breast 4T1 cancer cells[16,33], mouse melanoma B16F10 cells[16,33], human neurofibromatosis YMY-NF1 cells (from the same patient for the normal skin fibroblasts), human glioblastoma U87-MG cells, and mouse L929R fibroblasts. Available Supplementary Videos 13 to 39 for the Supplementary Fig. 2 are at https://doi.org/10.6084/m9. figshare.14560782, and video legends in the Description of Additional Supplementary Information.

**WWOXf cells repel WWOXd cells to undergo retrograde migration, and WWOXd cells induce apoptosis of WWOXf cells in return**. We scaled up the migration analysis using 18 additional pairs of WWOXd and WWOXf cells (or self-versus-self) under 2% FBS/medium (Supplementary Fig. 3a). In most cases, WWOXd cells, whether expressing WWOX or not, migrated individually and reluctantly toward WWOXf cells followed by retrograde migration. In return, WWOXd cells induced apoptosis of WWOXf cells (Supplementary Videos 1–3). An exception was that WWOXd glioblastoma U87-MG and 13-06-MG cells migrated forcefully toward L929S without retrograde migration, and then invaded the L929S cell area to cause apoptosis by >50% in 48 hr (Supplementary Fig. 3a10, 3a11, 4, and Supplementary Video 6). The protein expression profiles for WWOX, TβRII (type II TGFβ receptor), Hyal-2, ERα, β-actin, and α-tubulin in many tested WWOXd and WWOXf cells are shown by Western blots (Supplementary Fig. 3b, c).

**Increased WWOX isoform expression allows generation of aggressive behavior of tongue SCC cells**. When human oral SCC4, 9 and 15 cells were exposed to UV, all cells underwent BCD at room temperature (Supplementary Fig. 5a). These cells are WWOXf. The calcium influx was functional in SCC4 and 9. However, SCC15 had a reduced capability in calcium influx due to lack of wild type WWOX (46 kDa) (Supplementary Fig. 5b). In contrast to our previous report[42], SCC15 cells expressed isoform WWOX2 (41 kDa) and WWOX3 (30 kDa). Alteration in WWOX protein sizes is probably due to alternative splicing of mRNA or gene mutation during passages of cells in culture. UV-induced BCD was most effective in SCC9 and then SCC4, and SCC15 most resistant (Supplementary Fig. 5a). SCC15 aggressively migrated to SCC4 or SCC9 and pushed these cells to undergo retrograde migration (Supplementary Fig. 5c–f; Supplementary Video 7). L929S also forced HCT116 to move backward (Supplementary Fig. 3a15). Both cells are WWOXf. Overall, WWOX2 and 3 contribute, in part, to the increased aggressiveness of cell migration and induced calcium influx dysfunction. Available Supplementary Videos 40–42 for the Supplementary Fig. 5 are at https://doi.org/10.6084/m9.figshare.14560782.

**Sudden impact of WWOXd cell monolayers by WWOXf cells from suspension leads to activation of the ectopic survival IκBα/ERK/WWOX signaling in WWOXd cells**. We have identified a survival IκBα/ERK/WWOX signaling by tri-molecular time-lapse FRET microscopy[37]. MEF *Wwox* knockout cells were transiently overexpressed with ECFP-IκBα, EGFP-ERK, and DsRed-WWOX, then seeded in microtiter plates, and cultured for 24–48 hr. The knockout cells as a monolayer were impacted by adding the MEF wild type cells in suspension. By time-lapse triple protein-binding FRET microscopy, wild type cells induced the IκBα/ERK/WWOX signaling in the knockout cells. The data is shown as FRETc (relative FRET concentration)[37] in artificial white color and marked in red stars (Fig. 2a, b; Supplementary Videos 8, 9; see enlarged pictures in Supplementary Fig. 6a, b). Apoptosis occurred in the wild type cells (Fig. 2b; see white stars). Under similar conditions, when wild type cells were transiently overexpressed with expression constructs for the IκBα/ERK/ WWOX signaling, the knockout cells could not induce apoptosis

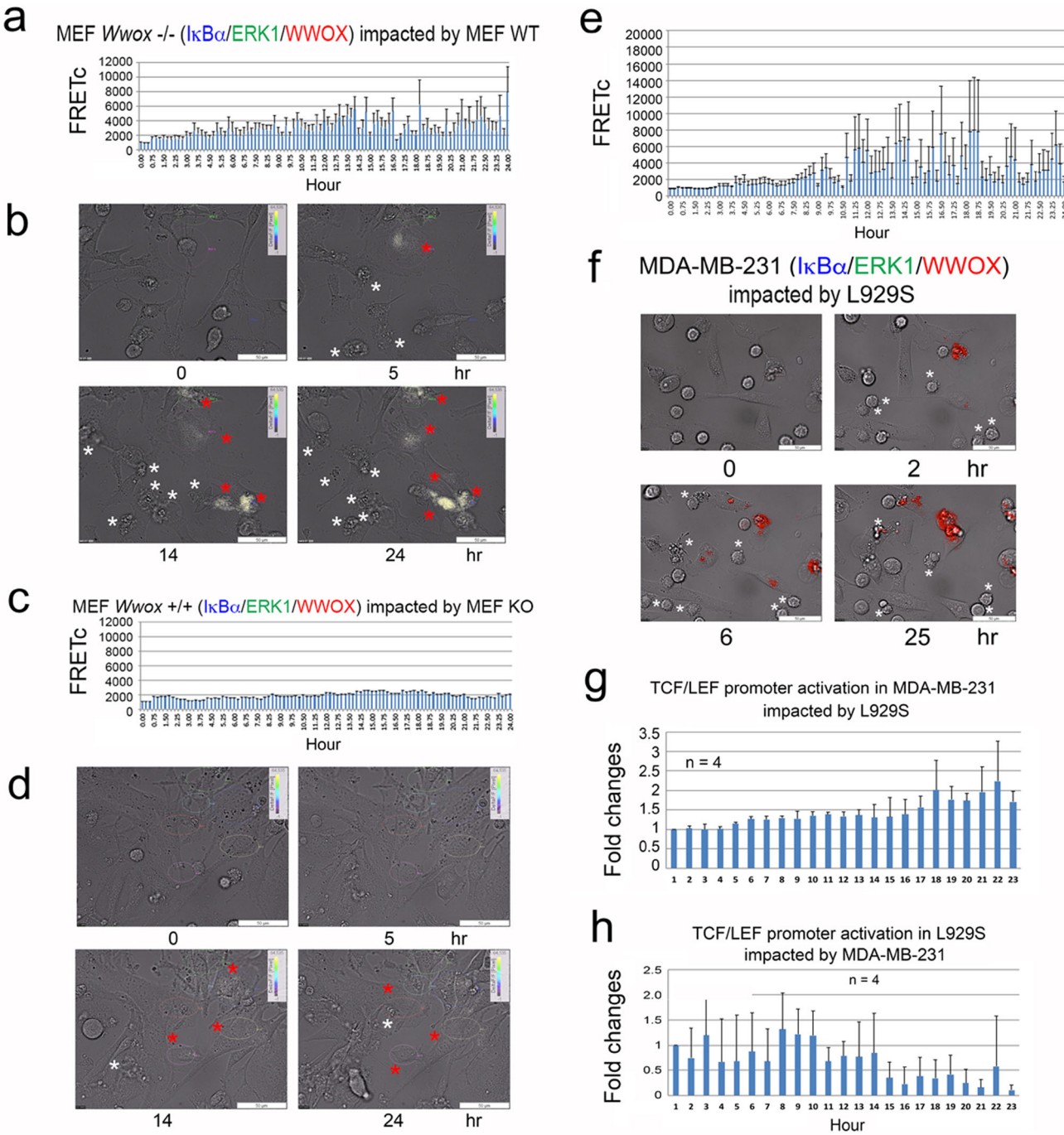

**Fig. 2 Sudden impact of WWOXd cells by WWOXf cells from suspension leads to activation of the survival IκBα/ERK/WWOX signaling in WWOXd cells, and WWOXf cells die. a, b** MEF *Wwox* knockout cells were transiently overexpressed with ECFP-IκBα, EGFP-ERK, and DsRed-WWOX, followed by culturing for 24–48 hr and adding MEF wild type cells from suspension for time-lapse FRET microscopy. Activation of the IκBα/ERK/WWOX survival signaling occurred in knockout cells (shown as FRETc in artificial white color and marked in red stars). Apoptosis occurred in the wild type cells (white stars). Also, see Supplementary Videos 10 and 11. **c, d** Upon sudden impact with the *Wwox* knockout cells from suspension, MEF wild type cells did not activate the ectopic IκBα/ERK/WWOX signaling complex (red stars). Knockout cells did not undergo apoptosis (white stars). Also, see Supplementary Videos 12 and 13. **e, f** By sudden impact with L929S from suspension, activation of the ectopic IκBα/ERK/WWOX signaling complex in breast MDA-MB-231 cells occurred (red). Apoptosis of L929S also occurred (white stars). **g** Upon sudden confrontation with L929S, MDA-MB-231 cells exhibited a gradual increase in the activation of TCF/LEF promoter, as determined by green fluorescence of time-lapse microscopy. **h** Under similar conditions, L929S exhibited a rapid increase in the activation of TCF/LEF promoter in response to the sudden impact by MDA-MB-231 cells from suspension.

of the wild type cells (Fig. 2c, d; Supplementary Videos 10, 11; Supplementary Fig. 6c, d for enlarged pictures).

Under similar conditions, MDA-MB-231 cells were transiently overexpressed with ECFP-IκBα, EGFP-ERK, and DsRed-WWOX and cultured for 24–48 hr. Sudden impact by non-transfected

L929S cells led to activation of the IκBα/ERK/WWOX signaling in MDA-MB-231 cells (shown as FRETc in red color), and meanwhile L929S cells died by apoptosis (Fig. 2e, f and Supplementary Video 12; Supplementary Fig. 6e, f for enlarged pictures).

**Sudden impact by L929S induces the late-phase activation of Wnt signaling in MDA-MB-231 cells**. When MDA-MB-231 cells aggressively migrate toward L929S, MDA-MB-231 cells exhibit a gradual increase in the Wnt signaling, as determined by the activation of the TCF/LEF promoter[38]. In contrast, no promoter activation of L929S is shown. Here, MDA-MB-231 cells were transiently overexpressed with a TCF/LEF promoter construct using GFP as a reporter[38]. Sudden impact of MDA-MB-231 monolayer by L929S from suspension resulted in gradual activation of the TCF/LEF promoter in MDA-MB-231, reaching ~100% increase in 18 hr (Fig. 2g). In contrast, sudden impact of L929S cells by MDA-MB-231 from suspension led to rapid increase in the activation of the TCF/LEF promoter in L929S within 14 hr (Fig. 2h), suggesting that TCF/LEF promoter activation is associated with MDA-MB-231-mediated apoptosis of L929S cells.

**TβRII in MEF wild type cells is responsible for repelling *Wwox* knockout MEF cells**. TGF-β1 induces WWOXd cells to merge with the WWOXf cells without causing apoptosis[38]. To determine the role of TβRII, wild type $Wwox^{+/+}$ MEF cells were pretreated with anti-TβRII IgG (2 µg/ml) for 24 hr, followed by gentle washing and processing migration assay versus knockout $Wwox^{-/-}$ MEF cells. Both wild type and knockout cells migrated aggressively in an anterograde manner and invaded each other's area, without causing apoptosis (Fig. 3a, c). In controls, wild type MEF cells remained relatively stationary as they stayed within <100 µm of migration distance (Fig. 3b). The accelerated migration of wild type cells was most pronounced when both MEF wild type and *Wwox* knockout cells were exposed to TβRII IgG during migration (Fig. 3d), or when the *Wwox* knockout cells were pretreated with TβRII IgG for 24 hr prior to processing the migration assay (Fig. 3a, e). The accumulated migration distance for the aforementioned experiments is shown (Supplementary Fig. 7).

In comparison, when L929S cells were pretreated with TβRII IgG (2 µg/ml) for 24 hr, these cells became largely stationary and were accessible to merge with MDA-MB-231 (Supplementary Fig. 8a). In stark contrast, when MDA-MB-231 were pretreated with TβRII IgG for 24 hr, both cells became relatively stationary (Supplementary Fig. 8a). Together, TβRII IgG treated-WWOXd cells become stationary upon facing untreated WWOXf cells.

**WWOXf cells induce apoptosis of WWOXd cells pretreated with pY33-WWOX or TβRII antibody without physical contacts**. WWOXd cells caused apoptosis of WWOXf cells from a distance of <500 µm (Fig. 1d, h, i, and 3f, g). When *Wwox* knockout MEF cells were pretreated with TβRII IgG for 24 hr, these cells underwent apoptosis caused by wild type MEF cells from a distance (Fig. 3f). In controls, knockout MEF cells did not induce apoptosis of TβRII IgG-treated knockout MEF cells (Fig. 3f). Similarly, pY33-WWOX antibody-pretreated L929R cells underwent apoptosis upon facing L929S cells (Fig. 3f). In controls, pY33-WWOX antibody-pretreated L929R cells did not undergo apoptosis caused by control L929R cells.

Also, TβRII IgG-pretreated MDA-MB-231 cells underwent apoptosis upon facing control L929R or L929S (Supplementary Fig. 8b). In contrast, TβRII IgG-pretreated L929R or L929S cells did not undergo apoptosis by encountering MDA-MB-231 cells (Supplementary Fig. 8b). Similarly, pY33-WWOX antibody-treated MDA-MB-231 cells underwent apoptosis upon facing untreated L929S, or pY33-WWOX antibody-treated L929S cells underwent apoptosis versus control MDA-MB-231 (Supplementary Fig. 8c). No cell death was observed when L929R and MDA-MB-231 were treated with or without pY33-WWOX antibody (Supplementary Fig. 8c). In control experiments, diluted normal

rabbit serum was used, which was relatively ineffective (Supplementary Fig. 8d). Shown in a schematic graph is a summary for L929S-mediated apoptosis of MDA-MB-231 pretreated with TβRII antibody (Fig. 3g). Additional combinations are also shown (Fig. 3g). Taken together, WWOXf cells kill WWOXd cells when these cells lose their endogenous mutant WWOX or TβRII.

**Identification of the complex of endogenous WWOX, Hyal-2, and TβRII in the lipid raft in vivo**. From the aforementioned observations (Fig. 3; Supplementary Fig. 8), we examined the presence of an endogenous complex of WWOX and TβRII. Membrane-bound WWOX possesses two cell surface epitopes, namely amino acid #7–21 (WWOX7-21) and #286–299 (WWOX286-299)[16]. When live MEF wild type cells were incubated with WWOX7-21 or WWOX286-299 peptide at 4 °C for 30 min followed by washing and processing immunostaining, these peptides colocalized with TβRII on the cell surface (Fig. 4a). In negative controls, no signals were observed without adding secondary antibodies.

Next, COS7 fibroblasts were transfected various expression constructs for EGFP-WWOX, EGFP-WW (possessing 2 N-terminal WW domains), EGFP-D3 (C-terminal tail), and EGFP-ΔD3WWOX. By colocalization analysis, both C-terminal SDR domain and D3 tail were needed for the localization of WWOX in the lipid raft (Fig. 4b). The N-terminal WW domain did not support localization of WWOX in the lipid raft (Fig. 4b). When cells were treated with UV irradiation (480 mJoule/cm²) followed by resting at 37 °C for 30 min, localization of WWOX in the lipid raft was not abolished (Fig. 4b).

By co-immunoprecipitation using specific WWOX antibodies[34–37], endogenous pY287-WWOX physically bound TβRII and Hyal-2 in the lipid raft in the liver cells of 3-month-old male wild type and *Wwox* heterozygous mice (Fig. 4c). Flotillin is a marker protein for lipid raft. pY33- and pS14-WWOX were less effective in binding TβRII (Fig. 4c). Endogenous Zfra, a zinc finger-like protein[15–17], effectively bound the cytosolic WWOX/TβRII complex of liver cells of BALB/c mice under different treatments (Fig. 4d). Liver samples were from our recent experiments for tumor growth in mice[16]. BALB/c mice received Zfra4-10 and/or WWOX7-21 peptide injections once per week for 3 consecutive weeks. One week later, all mice received inoculations of mouse breast cancer 4T1 cells and were then sacrificed 2 months later. Organs were harvested for processing co-immunoprecipitation. Compared to controls, Zfra4-10 or WWOX7-21 peptide significantly suppressed cancer growth[16]. In combination, both peptides enhanced cancer growth[16]. In addition, BALB/c mice received Zfra4-10 and/or WWOX7-21, or WWOX7-11 peptide injections for similar experiments. Presence of the pY33-WWOX/Hyal-2/TβRII complex was shown in the spleen (Fig. 4e). Full-length gels for Fig. 4c–e are shown in the section B of Supplementary Figures of Supplementary Information.

**WWOX7-11 peptide blocks WWOXd killing of WWOXf cells**. WWOX possesses two N-terminal WW domains, a nuclear localization sequence (NLS) between the WW domains, a C-terminal short-chain alcohol dehydrogenase/reductase (SDR) domain, and a proapoptotic D3 tail[1–4,16,38] (Fig. 5a). WWOX7-21 and WWOX7-11 sequences are located in front of the first WW domain of WWOX (Fig. 5a). From a remote distance of 500 µm, knockout *Wwox* MEF cells dramatically increased the redox activity (>5-fold; stained by RedoxSensor Red CC-1[38]) in the wild type cells (Fig. 5b). This led to apoptosis of wild type cells (Fig. 3f). Non-specific uptake of CC-1 by cells was shown at time zero. When WWOX7-11 peptide (10 µM) was used to pretreat

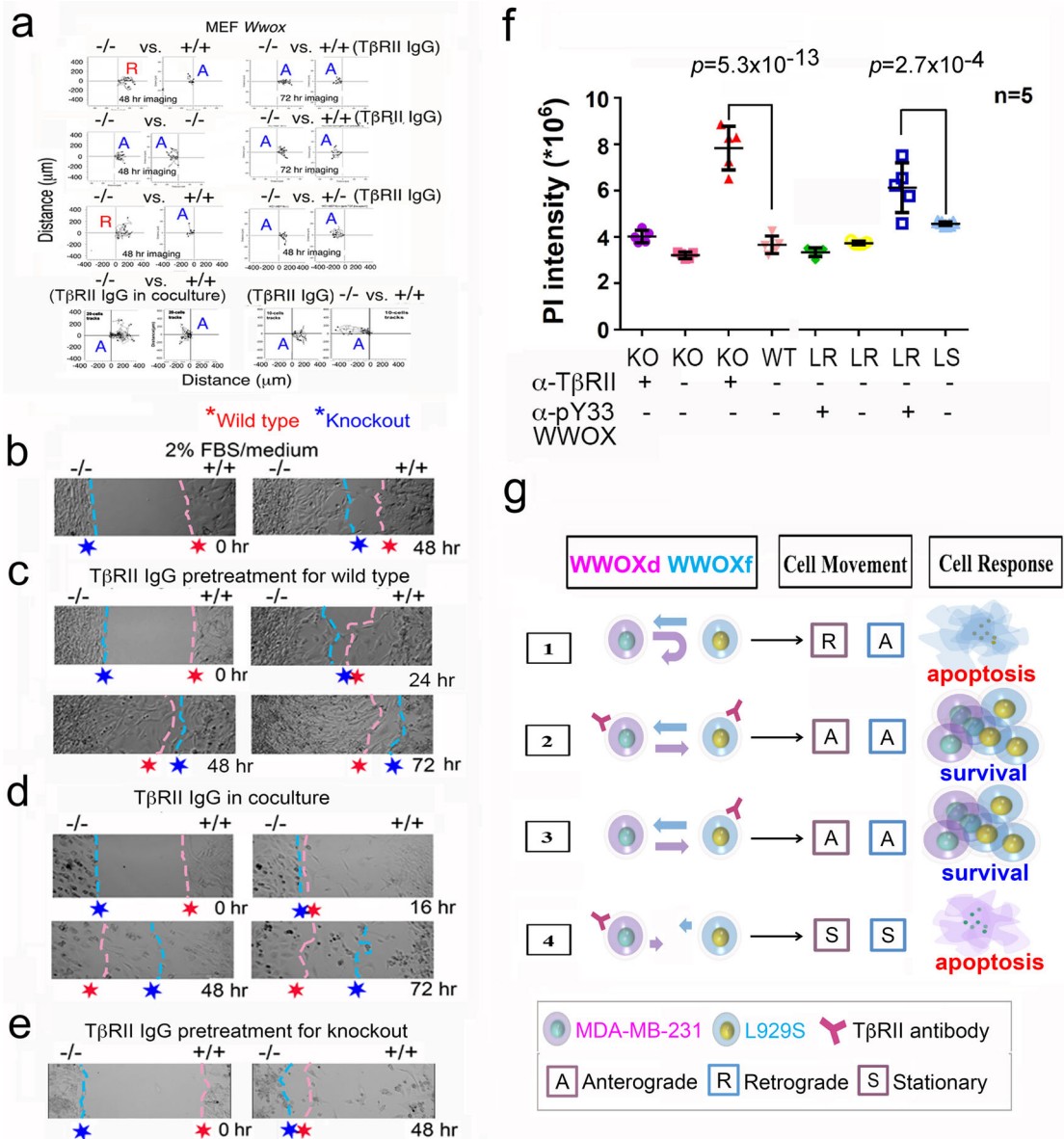

**Fig. 3 TβRII controls cell-to-cell migration. a**, **b** Migration of MEF *Wwox*⁻/⁻ versus *Wwox*⁺/⁺ or *Wwox*⁻/⁻ cells (randomly tracking 10 cells; in duplicates). The knockout cells underwent retrograde migration upon encountering the wild type cells. **c** Pretreatment of wild type cells with TβRII IgG (2 µg/ml) for 24 hr allows both cells to merge. **d** MEF wild type and *Wwox* knockout cells were co-treated with TβRII IgG during cell migration. Both cells merged and the wild type cell migration was accelerated. **e** When knockout *Wwox* cells were pretreated with TβRII IgG, the cell migration was retarded. **f** MEF *Wwox* knockout cells were pretreated with TβRII IgG (2 µg/ml) for 24 hr, followed by washing and processing the migration assay versus knockout cells or wild type MEF cells for 24 hr. Similarly, L929R cells were pretreated with pY33-WWOX antiserum (1:100 dilution) for 24 hr, and then subjected to the migration assay versus L929R or L929S cells. The extent of cell death was measured by PI uptake. **g** A summarized schematic graph illustrates the role of TβRII in regulating the migration of WWOXd versus WWOXf cells.

MEF wild type or *Wwox* knockout cells for 2 hr prior to washing and running the migration assay, the increased redox activity in wild type cells was abolished (Fig. 5c, d). No cell death was observed. In controls, WWOX7-11 did not induce redox activity when wild type versus wild type cells (Fig. 5e). The observations suggest that WWOX7-11 peptide supports cell survival via blocking increased redox activity.

**The *N*-terminal WWOX7-21 "gre" epitope greets visiting cells, and the *C*-terminal WWOX286-299 "repl" epitope repels WWOXd cells.** When WWOX286-299 peptide (200 µM) was coated onto the culture dish surface, MDA-MB-231 cells strongly

resisted to migrate over to the peptide-coated area (Fig. 5f). The WWOX286-299 peptide is hereafter designated as "repl" for its ability to repel visiting cells. When WWOX286-299 is phosphorylated at Y287, the repellence activity of the phosphorylated peptide was significantly reduced (~50% reduction; Fig. 5f; also see Fig. 5f_statistics in the section C of Supplementary Figures of Supplementary Information).

WWOX7-21 and WWOX7-11 peptides are potent in blocking cancer growth in mice[33]. In contrast, pS14-WWOX7-21 peptide dramatically increases cancer cell growth in vivo[33]. WWOX7-21 peptide is designated "gre" for its greeting or enhancing the migration of MDA-MB-231 cells toward the peptide-coated area (Fig. 5f). However, the S14-phosphorylated gre peptide,

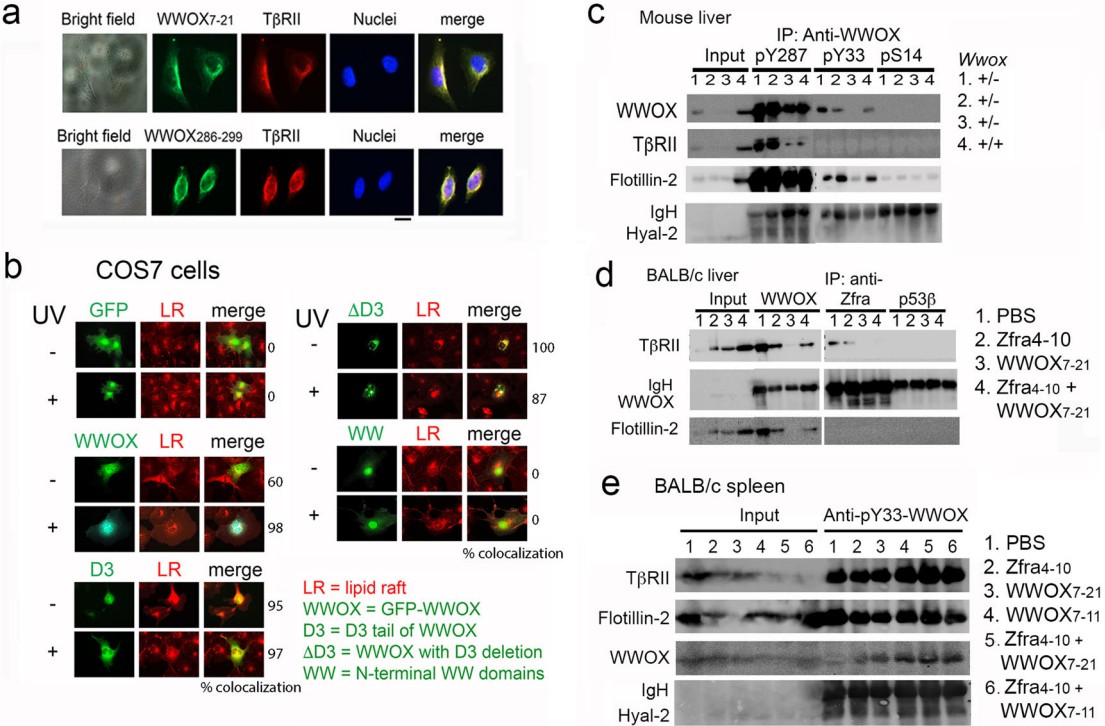

**Fig. 4 Identification of membrane WWOX/TβRII/Hyal-2 complex in the lipid raft in vivo. a** MEF wild type cells were incubated with WWOX7-21 or WWOX286-299 peptide at 4 °C for 30 min, followed by processing immunostaining for the peptides. TβRII colocalized with both peptides on the cell surface. **b** Colocalization analysis revealed that the C-terminal SDR domain and D3 region are needed for localization in the lipid raft (stained with anti-flotillin-2 IgG). **c** By co-immunoprecipitation, pY287-WWOX bound TβRII and Hyal-2 in the lipid raft of liver of Wwox wild type and heterozygous mice. pY33- and pS14-WWOX were less effective. Zfra also bound the WWOX/TβRII complex, not localized in the lipid raft. p53β bound TβRII only. **d** BALB/c mice received Zfra4-10 and/or WWOX7-21 peptide injections once per week for 3 consecutive weeks. After 1 week, all mice receive inoculations of mouse breast cancer 4T1 cells and sacrificed 2 months later. By co-immunoprecipitation, the WWOX/TβRII complex was shown in the lipid raft of mouse liver cells. **e** Similarly, BALB/c mice received the indicated peptide combinations, followed by inoculation with 4T1 cells and sacrificed 2 months later. Presence of the WWOX/TβRII/Hyal-2 was found in the lipid raft.

designated pS14gre or pS14-WWOX7-21, lost its enhancing activity for cell migration as compared to controls (Fig. 5f). WWOX11-20, without amino acid AGLD (#7–10), had no effect on cell migration (Fig. 5f). The first WW domain peptide "WWOX28-42" and its phosphorylation at Y33 (pY33-WWOX28-42) had no effect (Fig. 5f). Phosphorylation at Tyr33 is known for WWOX activation to block cancer growth[4,30,34–36]. The unique C-terminus of WWOX2 (also known as WOX2[35,43]) at amino acid #353–363 had no effect (Fig. 5f). In additional controls, scrambled peptides were made, i.e., gre (7–21scrm) and repl (286–299scrm), and both peptides had no effect on cell migration (Fig. 5f), indicating the specific activity of gre and repl peptides.

We quantified the extent of peptide coating onto the plastic surface by ELISA. For example, using 15 ng of a 13–15-amino-acid peptide (in 40 μl MilliQ or ~400 nM) for coating onto plastic surface, or serum or albumin-pretreated plastic surface. All peptides were coated with a relatively even amount, except that the amount of the coated pY287repl was doubled (Supplementary Fig. 9a, b). This is due, in part, to the unique composition of repl and pY287repl. Coating efficiency analysis revealed that when high levels of peptides were used, the coating efficiency was reduced (Supplementary Fig. 9c), suggesting that saturation of peptide binding onto the plastic surface has reached.

**Gre or repl peptide-coated cells affect the migration pattern of visiting cells.** When MDA-MB-231 cells were pretreated with gre or repl peptide (20 μM) for 4 hr at 37 °C and then washed to

remove the unbound peptides, control MDA-MB-231 cells exhibited an enhanced migration toward the gre-treated MDA-MB-231 cells (Fig. 5g, i). However, MDA-MB-231 underwent retrograde migration upon facing repl-treated MDA-MB-231 cells (Fig. 5g, j). In comparison, both control cells migrated in an anterograde manner to merge with each other (Fig. 5g, h). These synthetic peptides were not cytotoxic to cells in vitro, as determined by cell cycle analysis. Also, the peptides are not toxic to mice[33]. In additional peptide-coating experiments, WWOX7-11 and WWOX7-11A7R strongly attracted U87-MG cells to migrate forward to the peptide-coated area, whereas WWOX7-11G8R had no effect (Supplementary Fig. 9d). The observations suggest a critical role of Gly8 in supporting the enhancing activity of the gre (or WWOX7-21) epitope. Also, repl peptide inhibited the anterograde migration of U87-MG cells, and pY287repl did not have an inhibitory activity (Supplementary Fig. 9e). Overall, phosphorylation of Y287 in repl of WWOX significantly increases its binding to the plastic surface coated with or without albumin or serum proteins, and this reduces repl's repellence activity.

**Specific antibody blocks gre or repl epitope-regulated cell–cell recognition and migration.** We generated antibodies against gre, pS14gre, repl, and pY287repl peptides, respectively[37]. Neutralization of the coated gre peptide by a specific antibody abolished the enhancement of MDA-MB-231 cell migration (Fig. 6a; also see Fig. 6a_statistics). Non-immune serum had no neutralizing effects (Fig. 6a). Similarly, the migration inhibitory effect of the coated repl peptide was abolished by a specific antibody

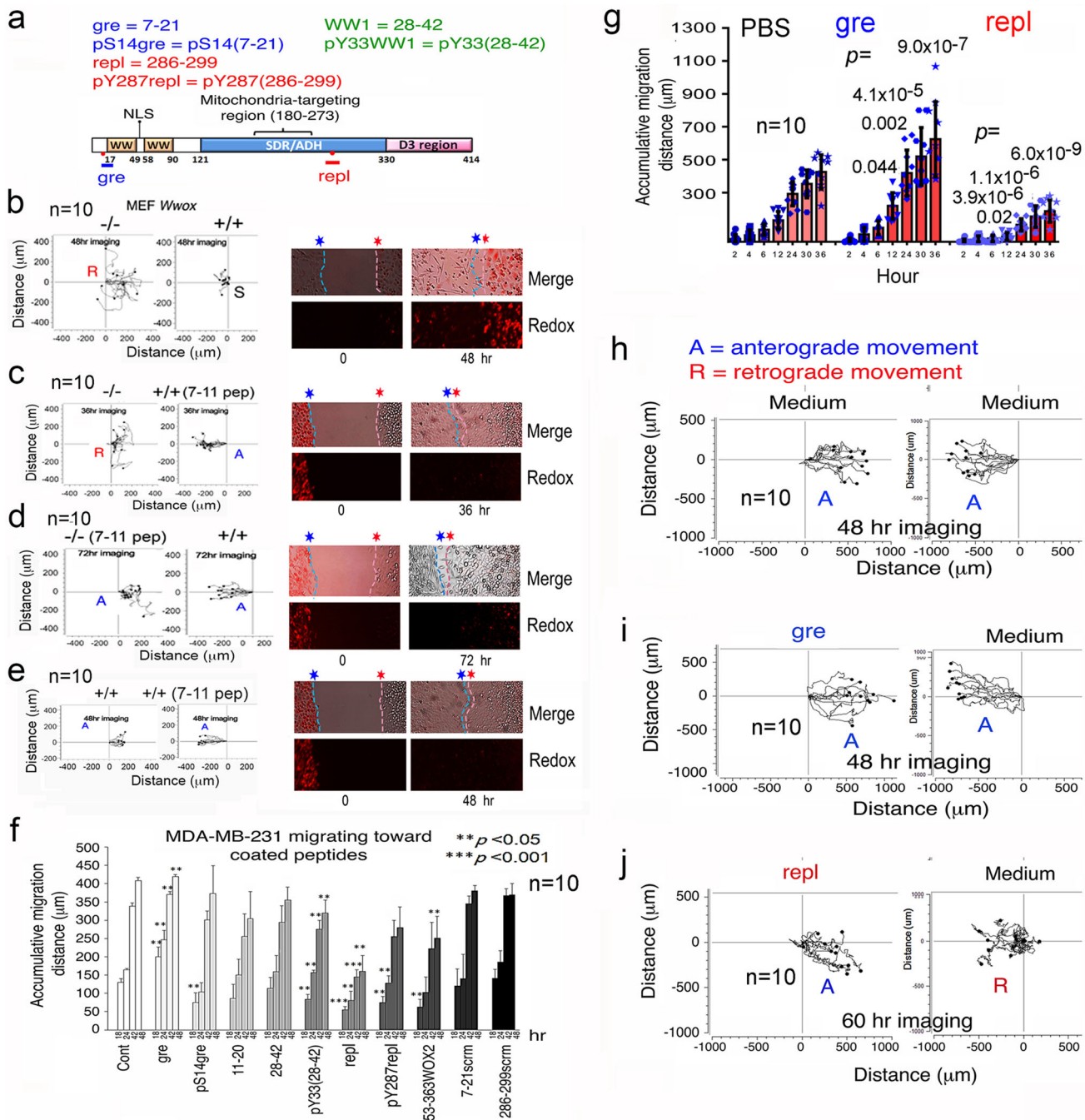

**Fig. 5 Gre (WWOX7-21) peptide enhances cell migration whereas repl (WWOX286-299) peptide retards cell migration. a** The primary structure of WWOX and localization of synthetic peptides are shown. **b–e** WWOX7-11 (10 µM) or PBS control was added to one of the MEF wild type versus MEF *Wwox* knockout cell pair for migration by time-lapse microscopy. The redox sensor CC1 was added to the cells at the left chamber, followed by starting the assay. Knockout cells induced redox activity in the wild type cells (**b**), as determined by CC1 incorporation. The induction was blocked by WWOX7-11 peptide when either the wild type or knockout cells received the peptide treatment. **f** One chamber of the culture-insert (ibidi) was coated with 200 µM (~15 ng) gre or repl peptide (or conjugation buffer only) overnight prior to washing, and the other side seeded with cells. Migration of MDA-MB-231 cells to each indicated peptide was carried out. Accumulative travel distance with time is shown. **g** MDA-MB-231 cells were seeded in the left chamber and pretreated with gre or repl peptide (20 µM) for 4 hr at 37 °C, followed by washing to remove the peptides. Untreated control cells were seeded at the right chamber. The accumulative distance of cell migration with time is shown. **h–j** Similarly, by time-lapse microscopy, gre peptide-treated MDA-MB-231 cells enhanced the migration of untreated control cells leading to merge. In contrast, repl peptide-treated MDA-MB-231 cells induced retrograde migration of the control cells. Tracking the migration of 10 randomly selected cells is shown.

(Fig. 6b; also see Fig. 6b_statistics in the section C of Supplementary Figures of Supplementary Information). Non-immune serum and antiserum against pY287repl had no neutralizing effects (Fig. 6b). Tyr287 is a conserved phosphorylation site,

which is associated with WWOX degradation via ubiquitination and proteasome[44].

To validate the antibody specificity, we carried out peptide blocking for the produced antisera[28]. Pre-incubation of gre

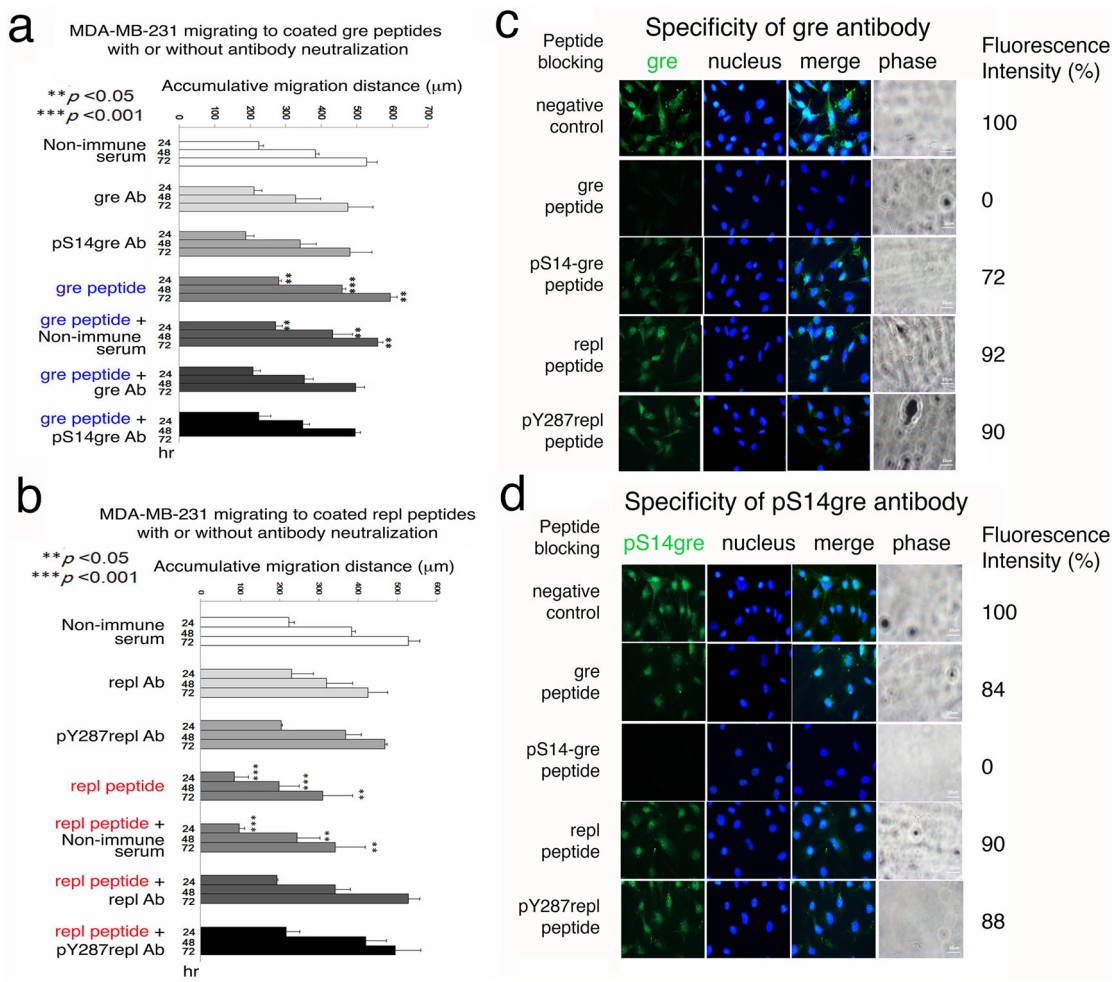

**Fig. 6 Neutralization of gre and repl peptides in regulating cell migration by specific antibodies. a, b** Gre and pS14gre peptides (200 μM) were coated onto the plastic surface, followed by washing and then treating with or without aliquots of diluted antisera (1:500), or non-immune sera for controls. Migration of MDA-MB-231 cells to each coated peptide area was imaged at indicated times. Student's *t* tests were carried out for all experiments versus controls (mean ± standard deviation; *n* = 3). **c, d** 3 × 10⁵ B16F10 cells were seeded onto cover glasses overnight. Cells were fixed with 4% paraformaldehyde and permeabilized with 0.25% triton-X PBS. Prior to immunostaining, primary antibodies were pre-adsorbed with or without 1 mM gre, pS14gre, repl, or pY287repl peptides for 1 hr at room temperature.

peptide with an aliquot of specific antiserum for 30 min at room temperature completely abolished the immunoreactivity in melanoma B16F10 cells (Fig. 6c). B16F10 cells, a WWOXd cell line, express WWOX. Antiserum against pS14gre, repl, or pY287repl did not block the immunogenicity of gre peptide (Fig. 6c). Similarly, the immunogenicity of pS14gre, repl, or pY287repl was blocked by the corresponding specific peptide only (Fig. 6d and Supplementary Fig. 10a, b).

**SDR domain possesses an oxidoreductase activity and repels visiting WWOXd cells.** We established stable transfectants of MDA-MB-231 cells expressing EGFP-tagged SDR domain or EGFP only (Fig. 7a; Supplementary Fig. 11a–e; Supplementary Table 1, exp 23–28, blue block). The extent of ectopic expression is shown (Fig. 7a). The ectopic SDR domain exhibited a strong redox activity (stained with CC-1; Fig. 7b; Supplementary Fig. 11a, b, e). Again, the ectopic SDR domain was localized in the lipid raft (Fig. 7b; Supplementary Fig. 11a). The SDR domain is Y287 phosphorylated (Supplementary Fig. 11c). The immunizing peptide for antibody production blocked the immunofluorescence (Supplementary Fig. 11d).

We continued to investigate the role of SDR domain. When control MDA-MB-231 cells encountered EGFP-expressing MDA-MB-231 cells, both cell groups migrated in an anterograde manner and merged eventually (Fig. 7c). However, EGFP-SDR-expressing cells caused the control cells to undergo retrograde migration (Fig. 7d). An aliquot of non-immune serum did not abolish the retrograde migration (Fig. 7e). Inhibition of the cell surface-exposed SDR domain by repl antibody led the EGFP-SDR-expressing cells to undergo anterograde migration and eventually merge with the control cells (Fig. 7f). The repl is within the SDR domain (Fig. 5a). Stimulation of EGFP-SDR cells with pY287-WWOX (or pY287repl) antibody could not convert the retrograde into anterograde migration in control cells (Fig. 7g), suggesting that Y287 phosphorylation is not involved in the cell repellence. EGFP-SDR-expressing cells exhibited significantly reduced migration velocity and migratory distance upon facing control MDA-MB-231 cells (Fig. 7h, i). In addition, EGFP-gre (WWOX7-21)-expressing MDA-MB-231 cells were established. These cells greeted control MDA-MB-231 cells. Gre antibody could not abolish the cell-to-cell greeting (Supplementary Table 1, blue box, exp 27 and 28). The aforementioned data are summarized in the Supplementary Table 1, blue box for exp

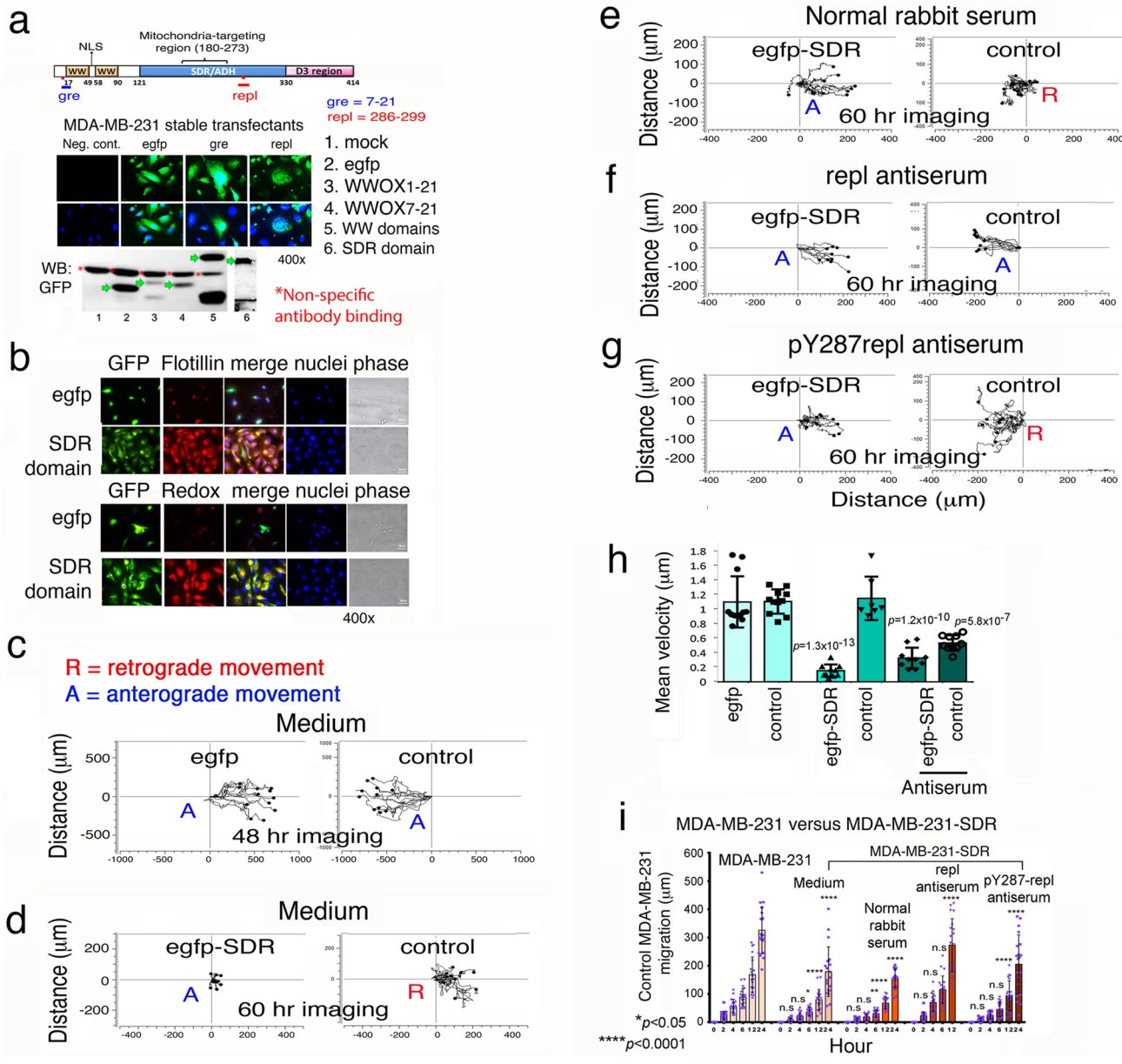

**Fig. 7 SDR domain is responsible for repelling cells. a** Stable transfectants of MDA-MB-231 cells with EGFP or EGFP-SDR domain expression were established. Immunofluorescent staining and Western blotting were performed to confirm the protein expression. The expressed GFP-containing proteins are marked by green arrows. A major non-specific protein recognized by the GFP antibody is marked with a red asterisk (*). **b** SDR-expressing MDA-MB-231 cells were seeded overnight, followed by processing immunostaining without cell permeabilization. The SDR domain colocalizes with Flotillin-2, which is a caveolae-associated, integral membrane protein. The SDR domain exhibits upregulated redox activity, as determined by RedoxSensor red CC-1 stain. **c–g** Stable transfectants of MDA-MB-231 cells with EGFP or EGFP-SDR domain (left) and MDA-MB-231 control cells (right) were seeded, respectively, in each side of the culture-insert (ibidi). After overnight incubation, time-lapse microscopy was carried out at 37 °C with 5% $CO_2$. Control cells migrated in a retrograde manner upon encountering the SDR domain-expressing cells. Treatment of SDR-expressing cells with antiserum against SDR domain (1:500 dilution) switched the retrograde migration of the control cells turn into anterograde migration. The pY287repl antiserum and control sera had no effect. **h** Mean velocity of each cell group is shown. Statistical analysis: egfp cell group versus each indicated cell group ($n = 10$). **i** Accumulative migration distance is shown for every cell group ($n = 10$; mean ± sd; One-way ANOVA).

23–28. Together, cell surface-exposed SDR domain in WWOX is responsible for repelling migrating WWOXd cells such as metastatic cancer cells. In other words, WWOXd metastatic cancer cells would face repellence in homing to a normal organ possessing WWOXf cells with membrane repl epitope.

**Repl antibody abolishes WWOXf cell-regulated retrograde migration of WWOXd cells.** When MDA-MB-231 cells encountered L929S cells, MDA-MB-231 underwent retrograde migration (Fig. 8a; Supplementary Fig. 3a1). Pretreatment of L929S cells with repl antibody for 1 hr at 37 °C resulted in anterograde movement of MDA-MB-231 cells toward the treated L929S cells (Fig. 8b). Also, we used velocity autocorrelation function (VACF) analysis[38,45] to determine the movement of MDA-MB-231 versus L929S cells (Fig. 8c–e). In the repl antiserum-treated L929S group, MDA-MB-231 migrated at a slower velocity than untreated control cells (Fig. 8c, d). Total

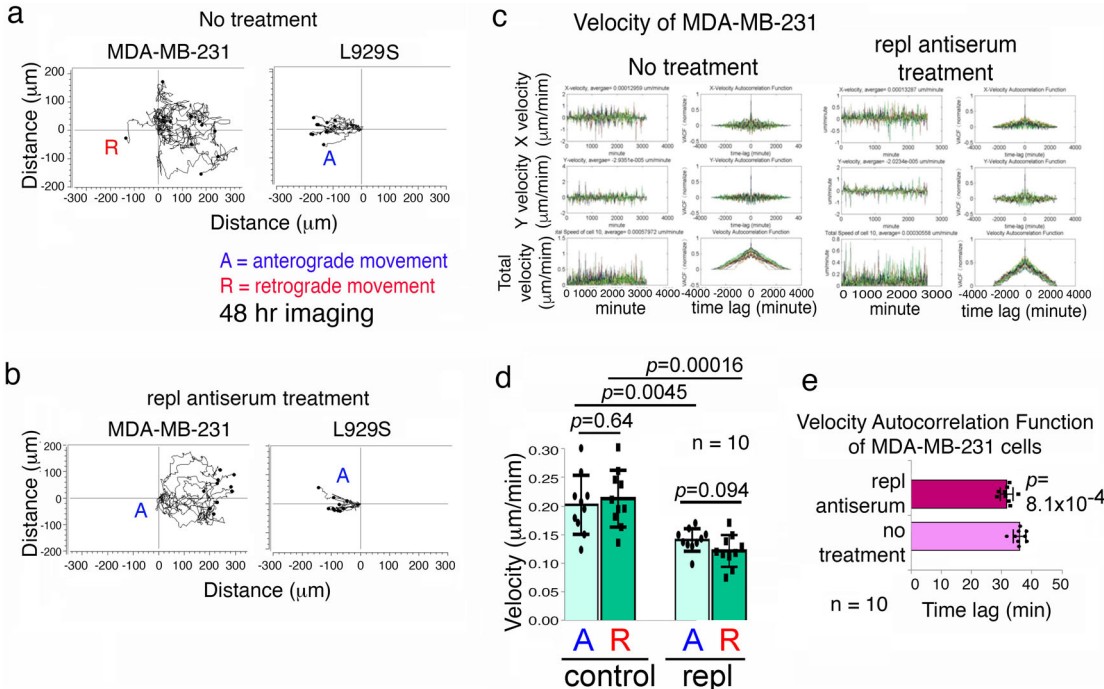

**Fig. 8 Repl antibody abolishes L929 cell-mediated retrograde movement of MDA-MB-231 cells.** MDA-MB-231 cells (left) and L929 cells (right) were seeded, respectively, in each side of the culture-insert (ibidi). After overnight incubation, time-lapse microscopy was carried out at 37 °C with 5% CO2. **a** WWOX-expressing L929 cells repelled the visiting WWOX-deficient MDA-MB-231 cells, which underwent retrograde migration. **b** Antiserum against SDR domain (1:500 dilution) abolished retrograde migration of MDA-MB-231 cells. **c** Temporal correlation of the velocity was performed to randomly analyze the migratory behavior of 10 cells. X, Y and total velocity were calculated[37]. The velocities of cells in each time points were shown as peaks. The velocity of MDA-MB-231 under repl antiserum treatment was lower than control. The total velocity autocorrelation function was also lower than control. With the treatment, MDA-MB-231 cells took less time to become uncorrelated to the initial condition. It reflected that repl antiserum treatment altered the movement of MDA-MB-231 and reduced the sensitivities of cells upon sensing foreign cells. **d** Velocities of anterograde and retrograde movements were shown. Repl Ab treatment also caused significant decrease of anterograde and retrograde movements. **e** The statistics and figures of total velocity autocorrelation function were shown. *A* anterograde movement, *R* retrograde movement. Each picture was taken per 10 min. (mean ± standard deviation; *n* = 10; Student's *t* test).

VACF for the treated cells was also significantly lower than controls (Fig. 8c, e). However, there were no differences between anterograde and retrograde movement for each cell group (Fig. 8d).

To simulate MDA-MB-231 cells encountering a WWOX-expressing tissue or organ, we utilized MDA-MB-231 and primary lung epithelial cells to perform time-lapse migration assay (Supplementary Fig. 12). In controls, anterograde movement was observed when MDA-MB-231 versus MDA-MB-231 cells (Supplementary Fig. 12a). As expected, MDA-MB-231 underwent retrograde movement upon encountering primary lung culture cells (Supplementary Fig. 12b). However, repl antiserum-pretreated primary lung culture cells induced anterograde migration of MDA-MB-231 cells, and both cells merged finally (Supplementary Fig. 12c). The accumulative cell migratory distance and mean velocity of all tested cells under different conditions were calculated (Supplementary Fig. 12d, e). The data showed that repl antiserum (or SDRrepl antibody) significantly increased the migration of the primary lung epithelial cells. Similar results were observed using VACF analysis[38,49] to determine the movement of MDA-MB-231 when encountering the primary lung culture cells (Supplementary Fig. 12f–h). In the repl antiserum-treated lung culture cell group, MDA-MB-231 migrated at a slower velocity than untreated control cells (Supplementary Fig. 12f–h). Total VACF for the treated cells was also significantly lower than the control cells (Supplementary Fig. 12f–h).

In the Supplementary Table 1, we summarized the results from the aforementioned experiments and added additional experiments (Exp. 29–48) using different cell pairs. For example, when the MDA-MB-231/L929S coculture was added repl antibody (1:100 dilution) or TGF-β1 (10 ng/ml), both cells migrated forward and then merged without causing apoptosis to each other (Exp. 14 and 18, purple block). When antibody against pY287-repl, gre or pS14-gre was added in the coculture, MDA-MB-231 underwent retrograde migration and apoptosis occurred mainly in L929S (Exp. 15–17, purple block). In controls (Exp. 12 and 13), MDA-MB-231 moved backward upon facing L929S, and L929S subjected to apoptosis. The data for MDA-MB-231-SDR versus MDA-MB-231 in the presence of specific antibodies is summarized (Fig. 7c–g; Exp. 23–26, blue block). Again, antibody against gre did not block the anterograde migration of MDA-MB-231-gre and MDA-MB-231 in coculture (Exp. 27 and 28, blue block). In parallel with the effect of WWOX7-11 peptide (Fig. 5b–e), when MDA-MB-231 cells were pretreated with gre or pS14-gre peptide, these treated cells and control cells underwent anterograde migration and then merged (Exp. 19 and 20). Repl peptide conferred repellence to the visiting control MDA-MB-231 cells (Exp. 21), whereas repl or gre peptide-treated MDA-MB-231 cells could not merge with each other (Exp. 22).

**Repl antibody and TGF-β1 suppress the SDR-mediated repellence to facilitate cancer cell invasion.** TβRII antibody enhanced the invasion of L929S to MDA-MB-231 cells, and MEF wild type

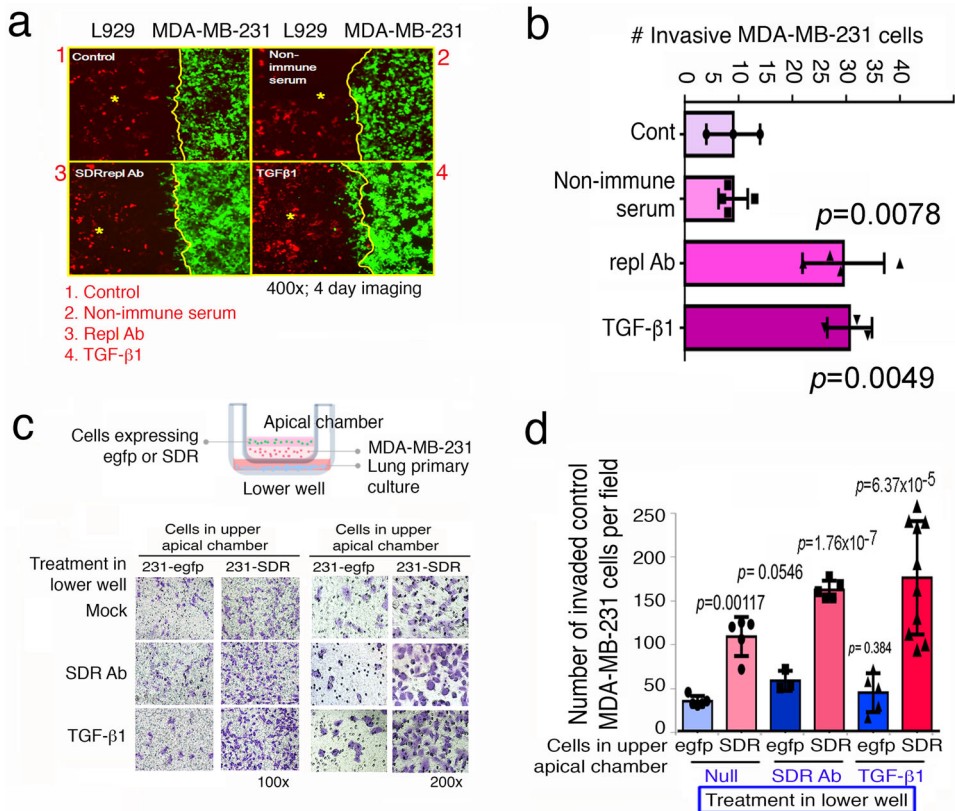

**Fig. 9 TGF-β1 utilize membrane TβRII and SDR domain of WWOX to regulate cell invasion. a, b** Stable transfectants of MDA-MB-231-EGFP and L929-DsRed2 were established. These cells were seeded in either side of the chambers of the culture-insert (ibidi), respectively. After overnight culture, the insert was gently removed. The cells were cultured at 37 °C with 5% $CO_2$ for 4 days. Image J software was used to count MDA-MB-231 cells invading into the L929 cell mass. Repl antiserum and TGFβ1 treatments significantly increased the number of invasive MDA-MB-231 cells. Non-immune serum treatment had no effect. (mean ± standard deviation; $n = 4$; Student's $t$ test). **c, d** Transwell assay was performed to determine the regulation of TGF-β1 and repl in cancer cell invasion. The experimental procedure is illustrated (see Supplementary Fig. 7). The invasive cells were stained with staining solution and performed with microscopy (Magnification 100X) (**c**). Parental MDA-MB-231, which were cocultured with cells-expressing SDR in apical chamber, had an increased invasive activity (**d**). Both repl antibody and TGFβ1 treatment in lower well also increased the number of invasive MDA-MB-231 cells (**d**). The 200X magnification was shown. (mean ± standard deviation; $n = 5$; Student's $t$ test).

cells to MEF *Wwox* knockout cells (Fig. 3c–e). To determine whether SDR domain regulates cell invasion, stable cell transfectants of MDA-MB-231 with EGFP and L929 cells with DsRed2 were established. Again, the cells were seeded onto either side of the chambers of a culture insert (ibidi), respectively. The cell migration event was allowed to last for 4 days in the presence of 1.5 ml RPMI/2% FBS (in 2 cm diameter dishes). Analysis by Image J software for cell counts revealed that pretreatment of L929 cells with repl antibody or TGF-β1 resulted in significant increases in the numbers of MDA-MB-231-EGFP cells in the L929 cell areas (Fig. 9a, b).

We performed Transwell assay to see the effect of SDR domain in forcing MDA-MB-231 cells to migrate through the matrix of gelatin gel. As shown in a schematic diagram (Supplementary Fig. 13), WWOXf lung primary cells were seeded in 0.1% gelatin onto the lower well to exert repellence activity. The apical chamber was seeded with control MDA-MB-231 cells in 0.1% gelatin to form a cell layer. Then, MDA-MB-231-EGFP or MDA-MB-231-EGFP-SDR in medium was layered onto the top of the gelatin-cell layer. When MDA-MB-231-EGFP-SDR cells were seeded over the top of control MDA-MB-231 cells in the apical chamber, control MDA-MB-231 cells acquired an enhanced invasion activity by ~2–3-fold (Fig. 9c, d). Under similar conditions, control MDA-MB-231-EGFP cells had no effect (Fig. 9c, d). The observations suggest that MDA-MB-231-EGFP-

SDR cells repel or force the control MDA-MB-231 cells to migrate through the gelatin matrix. When lung primary cells were treated with repl antibody or TGF-β1 in lower well to reduce the repellence activity, the control MDA-MB-231 cells in gelatin further gained their invasion capability by 50–75% (Fig. 9c, d).

**TGF-β1 enhances *Wwox*[+/+] MEF cell migration by upregulating stress fiber formation.** TGF-β suppresses epithelial cancer cell growth in the initiation stage. In contrast, metastatic cancer cells secrete a large amount of TGF-β to support their proliferation, invasiveness and metastasis[46–48]. Next, we determined the effect of TGF-β on cell migration and morphological changes. By using wild type and *Wwox* knockout MEF cells, TGF-β1 promoted the migration of wild type cells but suppressed the knockout cell migration in a dose-dependent manner (Fig. 10a). At 10 ng/ml, TGF-β1 and TGF-β2 promoted the migration of wild type MEF cells, whereas TGF-β2 suppressed the migration of knockout cells (Fig. 10b, c). Enhancement of wild type cell migration by TGF-β1 or TGF-β2 correlates with stress fiber formation as the F-actin appeared as bundles in cytoplasm (Fig. 10d). However, the effect was not observed in the knockout cells (Fig. 10d). Upregulation of stress fiber formation was further confirmed in TGF-β1-treated wild type MEF cells in a time-dependent manner (Fig. 10e).

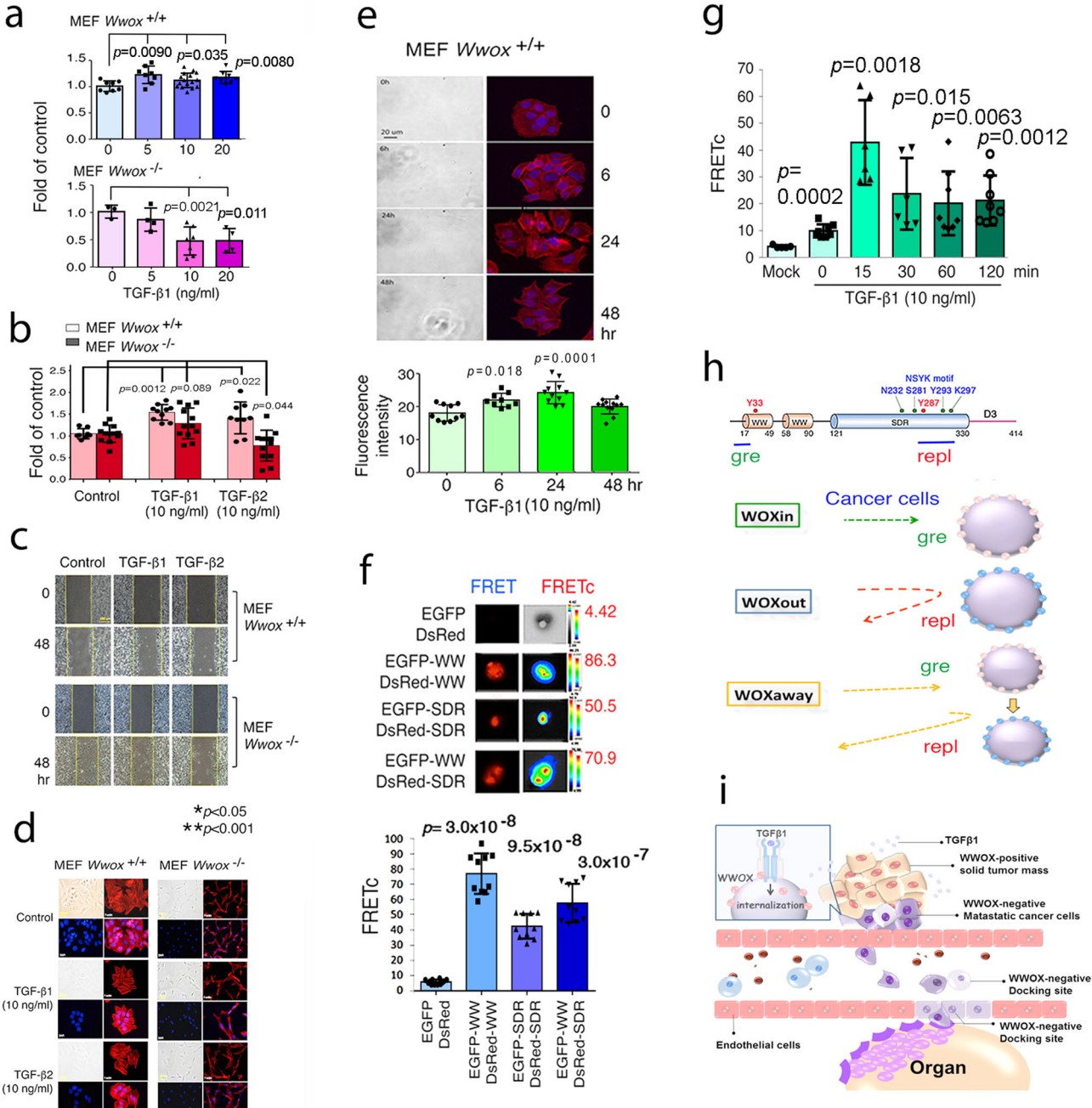

**Fig. 10 TGF-β1-enhancement of the migration of wild type MEF cell migration is associated with stress fiber formation and modulates WWOX self-binding or intermolecular association. a** *Wwox*⁺/⁺ and *Wwox*⁻/⁻ MEF cells were treated with TGF-β1 for 48 hrs. The extent of cell migration was measured. TGF-β1 promoted wild type cell migration and suppressed the knockout cell migration in a dose-dependent manner. **b**, **c** Under similar conditions, TGF-β1 and TGF-β2 promoted the wild type cell migration. TGF-β2 suppressed knockout cell migration. **d** MEF cells were treated with TGF-β1 or TGF-β2 for 48 h. In wild type cells, TGF-β1/-β2 induced stress fiber formation as the F-actin appeared as bundles in cytoplasm. However, in the knockout cells, the arrangement of F-actin showed no apparent differences between control and TGF-β stimulated cells. **e** In wild type MEF cells, TGF-β1 stimulated stress fibers formation in a time-dependent manner. F-actin (red) was stained by phalloidin. DAPI (blue) stain reveals the nuclei (mean ± s.d., *n* = 10, student's *t* test). **f** COS7 fibroblasts were co-transfected with a pair of expression constructs: (1) EGFP and DsRed, (2) EGFP-WW (*N*-terminal 1st and 2nd WW domains) and DsRed-WW; (3) EGFP-SDR and DsRed-SDR; (4) EGFP-WW and DsRed-SDR. FRET analysis was carried out. The extent of specific protein/protein interactions is calculated as FRETc. **g** TGF-β1 (10 ng/ml) increased the binding between WW and SDR domain in 15 min, followed by gradual decrease with time (mean ± standard deviation; *n* = 30; Student's *t* test). **h** The potential modes of cell migration are: (1) WOXin: WWOXd cells look for WWOXd or gre-exposed WWOXf cells in the lymphatic capillaries or organs for docking and invasion; (2) WOXout: WWOXd cells turn away from repl-exposed WWOXf cells; (3) WOXaway: WWOXd cells physically contact and merge with gre-expressing cells, but walk away upon gre turning to repl. **i** Cancer cell-derived TGF- β1 induces internalization of WWOX and TβRII that facilitate the cell migration. WWOXf parental cells in the solid tumor can no longer recognize the WWOXd cells and repel them to depart. These metastatic cells look for WWOXd or gre-exposed WWOXf cells in the blood or lymphatic capillaries as docking sites, so as to penetrate, dock and grow in target organs. TGFβ or probably other cytokines participate in WWOX-mediated cell migration and cell-to-cell recognition.

**WWOX self-binding via WW and SDR domains and TGF-β1 rapidly increases the binding followed by dissociation**. We investigated whether the *N*-terminal WW domain binds the *C*-terminal SDR domain intramolecularly, or the binding occurs at an intermolecular level. By Förster resonance energy transfer (FRET) analysis[7,30,32,33], we determined that WW domain physically bound SDR domain (Fig. 10f). Also, WW or SDR domain could undergo self-binding (Fig. 10f). Exogenous TGF-β1 rapidly increased the WW/SDR binding strength in 15 min, followed by gradual reduction (Fig. 10g).

**Cell-to-cell recognition and migration profiles**. We categorized the behavior of cell-to-cell migration as follows: (1) WOXin: WWOXd cells are ready to merge with another WWOXd cells or gre-exposed WWOXf cells in the lymphatic or blood capillaries or organs, which leads to docking, invasion, and homing; (2) WOXout: When WWOXd faces WWOXf, WWOXf forces WWOXd to undergo retrograde migration and WWOXd secrete cytotoxic molecules to kill WWOXf for return; (3) WOXaway: WWOXd physically contacts with gre-expressing cells, but moves away rapidly upon gre turning into repl (Fig. 10h). In the WOXin mode, WWOXd cells secrete TGF-β to overcome the repellence exerted by wild type WWOXf cells (Fig. 10i)[38]. Cancer cell-derived TGF-β1 induces internalization of WWOX and TβRII in the WWOXf cells that temporarily renders the cells to turn into WWOXd, which are readily susceptible to docking and homing by invading WWOXd cells. Together, our data support the scenario that parental WWOXf cells force WWOXd daughter cells to leave the home base. The metastatic daughter cells seek WWOXd cells in the lymphatics for penetration and relocation to another organ. These metastatic cells secrete TGF-β1 to compromise with WWOXf cells in a new organ for docking and homing (Fig. 10i)[38].

## Discussion

In summary, we have developed new methods to distinguish the differences between WWOXf cells and WWOXd cells. For the WWOXf group, cells respond to stress stimulus-mediated calcium influx and non-apoptotic BCD at room temperature and repel visiting WWOXd cells from a distance in coculture at 37 °C. Functional WWOX protein expression in normal cells (e.g., primary human skin fibroblasts) may not be abundant, but the protein expression is inducible by stress stimuli. Ceritinib and other therapeutic chemicals induce BCD at 37 °C[32,33]. In the WWOXd group, cells undergo explosion upon exposure to UV, UV/cold shock or chemotherapeutic chemicals at room temperature. Stress-regulated calcium influx does not work effectively in WWOXd cells. Also, WWOXd cells undergo retrograde migration upon encountering WWOXf cells. WWOXd cells may express abundant mutant WWOX protein or nothing at all. In agreement with our previous report[38], when WWOXf and WWOXd cells are set for migration to face each other from a distance of 500 μm, they sense each other rapidly. WWOXd cells significantly upregulate the redox activity of WWOXf cells and induce these cells to undergo apoptosis[38]. Meanwhile, WWOXd cells may undergo retrograde migration to avoid physical contacts with WWOXf cells.

In parallel, when adherent WWOXd cells (e.g., MDA-MB-231) face sudden impact by WWOXf cells (e.g., L929S) from suspension, WWOXd cells survive due to activation of the prosurvival IκBα/ERK/WWOX signaling[37]. This event does not occur in WWOXf cells. Alternatively, when adherent WWOXf cells are impacted by WWOXd cells from suspension, WWOXf cells cannot initiate the IκBα/ERK/WWOX signaling and undergo apoptosis. Wnt signaling-mediated TCF/LEF promoter activation occurs gradually in MDA-MB-231 with time upon impact by L929S. However, a rapid Wnt signaling occurs rapidly in L929S upon impact by MDA-MB-231. Whether this correlates with apoptosis in L929S is unknown. Together, failure in initiating the survival signaling leads to apoptosis of WWOXf cells.

An intriguingly finding is that when WWOXd cells are pre-treated with antibody against TβRII or pY33-WWOX, WWOXf cells induce apoptosis of WWOXd cells. Further, in most cases, both WWOXf and the TβRII or pY33-WWOX-treated WWOXd cells are stationary. TGF-β and TβRII are involved in metastasis of breast and other types of cancer cells[49–51]. Concomitant loss of WWOX and TβRII in WWOXd cells would result in abrogation of the TGFβ/TβRII/Smad signaling-mediated apoptosis. Thus, how WWOXf cells mediate apoptosis of WWOXd cells remains to be established.

When WWOXd cells sense the presence of WWOXf cells, WWOXd cells exhibit the activation of MIF, Hyal-2, Eph, and Wnt pathways, which converge to the MEK/ERK signaling[38]. These events enable WWOXd cells to dodge from the WWOXf cells. Inhibition of each pathway by antibody or specific chemicals in both WWOXf and WWOXd cells enables them to reconcile each other and merge[38]. Furthermore, WWOXd cells may compromise with the WWOXf cells by secreting TGF-β to acquire mutual recognition. That is, metastatic WWOXd cancer cells gain their advantage via TGF-β in successfully seeking and settling in a new home base. Mechanistically, TGF-β1 induces the internalization of membrane WWOX, Hyal-2[7] and TβRII in normal WWOXf cells. These cells temporally turn into a WWOXd phenotype, and thereby allow accessibility to the invasive WWOXd cells. the potential participating pathways are TGF-β/TβRII/Smads and Hyal-2/WWOX/Smad4[7,13,32].

The molecular mechanisms by which WWOXf cells undergo BCD and WWOXd cells explode in response to UV or UV/cold shock at room temperature are largely unknown. Suppression of membrane WWOX and TβRII by specific antibodies or chemical inhibitors prolongs the survival of WWOXf cells in response to stress stimuli. Calcium influx capability in these treated cells are also significantly suppressed, suggesting that both WWOX and TβRII are needed for UV-mediated BCD.

When antibody against Hyal-2 is added in the coculture, WWOXf and WWOXd undergo anterograde migration and merge[38]. Similar results are observed by adding TβRII or WWOX antibody in the coculture, supporting the notion that all components in the WWOX/Hyal-2/TβRII complex in the lipid raft are crucial in cell-to-cell sensing and recognition. By co-immunoprecipitation, we determined that WWOX, Hyal-2 and TβRII physically bind Flotillin-2 of the lipid raft in liver and spleen of mice. Other organs such as brain and small intestine also have the WWOX/Hyal-2/TβRII complex. WWOX localization in the lipid raft is via its *C*-terminal SDR domain and the D3 tail in binding Flotillin-2.

Loss of WWOX is frequently seen in metastatic cancer cells[1–6]. WWOX deficiency leads to the activation of JAK2/STAT3 signaling in metastatic triple negative breast cancer cells[52]. WWOX also regulates the expression of microRNA in the control of metastasis of triple negative breast cancer cells[53]. Our data support the scenario that WWOXf parental cells can no longer recognize the daughter WWOXd cells and repel them to depart from the original solid tumor or the home base. The metastatic cancer cells look for WWOXd cells in the blood or lymphatic capillaries as docking sites, so as to penetrate, dock and grow in target organs.

We determined by FRET microscopy that SDR domain binds another SDR domain or WW domain, and WW domain with another WW domain. Presumably, this results in the formation of a net-like network on the cell surface. This network interacts

with integrins, makes the cells less mobile, and renders them to undergo collective migration. From the repl peptide coating experiments, when repl is phosphorylated at Y287, there is an increased binding of the peptide to the extracellular matrix. Conceivably, the full-length protein pY287-WWOX has an increased affinity to extracellular matrix, so as to limit cell migration.

By using SCC cells, we demonstrated that SCC15 cells lose wild type WWOX but express WWOX2 and 3. Compared to SCC4 and 9, SCC15 cells have reduced cell sizes but migrate in a collective manner. Although all SCC cells undergo BCD in response to UV, calcium influx is less efficient in SCC15 than SCC4 and 9. Indeed, SCC cells are good models for determining cell migration and cell-to-cell recognition. WWOX2 and 3, which have altered SDR domains, contribute to enhanced cell migration, reduced calcium influx, increased cell migration and aggressiveness. SCC cells have provided a nice model for how loss of WWOX in rendering metastatic potential.

We have extensively examined the role of repl and gre epitopes in WWOX that affects cell-to-cell recognition and movement. WWOXf cells with surface exposure of the repl epitope strongly fends off WWOXd cells, whereas gre welcomes visiting cells without WWOX or expressing surface-exposed gre only. Conformational alterations of WWOX probably play a crucial role in determining cell-to-cell recognition. By FRET analysis, we demonstrated the potential of intramolecular folding via binding of N-terminal WW domain with C-terminal SDR domain—known as a "closed form". In most cases, when WWOXf cells are cultured in 10% FBS, Y33 is phosphorylated in WWOX. The C-terminal repl epitope in the SDR domain is concealed, and the N-terminal gre epitope is exposed for attracting cells. When WWOX is de-phosphorylation at Y33, its repl epitope is exposed onto the cell surface, which is considered as an "open form". Gre epitope is then hidden via binding WW domain or D3 tail, and the repl epitope exposed. Gre peptide undergoes self-polymerization, which correlates its activity in blocking cancer growth and metastasis in vivo[33]. Repl peptide may suppress cancer growth but does not block cancer metastasis in vivo[33]. Repl epitope in the SDR domain repels invading cancer cells, and whether this correlates with its self-polymerization is unknown. The molecular architecture of membrane WWOX has yet to be established by cryptography or cryo-electron microscopy.

TGF-β1 enhances the migration of WWOXf cells by upregulating stress fiber formation. Indeed, stress fiber orientation affects cell migration[54,55]. Wild type MEF cells migrate collectively and slowly. TGF-β1 and TβRII antibodies promote the migration of wild type cells by causing collective migration into individual migration, suggesting that the TGF-β1/TβRII/Smads or TGF-β1/Hyal-2/WWOX/TβRII/Smads pathway is involved. In contrast, TGF-β1 suppresses the migration of Wwox knockout cells and increases the cell migration in a collective manner, suggesting that TGF-β1-treated knockout cells secrete adhesion proteins that limit their migration. Compared to wild type MEF cells, knockout MEF cells have a significantly increased rate of proliferation[37]. Similar observations are also shown in skin keratinocytes[56]. TGF-β upregulates F-actin bundles in cytoplasm in a WWOX-dependent manner, suggesting that WWOX is needed for the F-actin bundle formation[57]. Upregulation of Rho GTPase activity has been shown at the leading edge of migrating cells.

In conclusion, membrane WWOX and TβRII participate in controlling cell-to-cell recognition and modes of migration (collective or individual). WWOXf cells repel the invading WWOXd cells, whereas WWOXd cells in turn induce apoptosis of WWOXf cells. The intensive cell-to-cell interaction is indeed a "bully" event among cells. We determined that repl epitope of WWOX is responsible for cell repellence. In contrast, the gre epitope is essential for WWOXf cell to greet and merge with the invading WWOXd cells. Metastatic cancer cells manipulate the expression of WWOX and TβRII to gain the advantage of survival.

## Methods

**Cell lines and primary cell cultures from organs.** Where indicated, all commercial cells were directly from American Type Culture Collections (ATCC). Cell lines used for the experiments are divided in 4 categories: (1) WWOXf cells, expressing functional WWOX protein, have been maintaining in 10% FBS/DMEM medium, including human breast cancer MCF7 cells, human colon cancer HCT116 cells, human prostate cancer DU145 cells, human neuroblastoma SH-SY5Ycells, human testicular cancer NT2D1 cells, monkey kidney COS7 fibroblasts, human non-small cell lung NCI-H1299 carcinoma cells, human neuroblastoma SH-SY5Y cells, human squamous cell carcinoma SCC 4, 9 and 15 cells, and human testicular cancer NT2D1 cells from ATCC; (2) WWOXd cells, expressing dysfunctional WWOX protein or none, have been maintaining in 10% FBS/DMEM medium, including human breast cancer MDA-MB-231 cells, human cancer MDA-MB-435s cells, murine breast cancer 4T1 cells, human neuroblastoma NB69 cells, murine melanoma B16F10 cells, and human glioblastoma U87-MG and 13-06-MG cells from ATCC; (3) WWOXf cells, cultured in 10% FBS/RPMI, were murine fibrosarcoma L929S, human normal skin fibroblasts, and mink lung epithelial Mv1Lu from ATCC, and Wwox wild type MEF from the embryos of B6 mice maintained in our laboratory; (4) WWOXd cells, cultured in 10% FBS/RPMI, were murine fibrosarcoma L929R, human neurofibromatosis NF1, and Wwox knockout MEF (from exon 1 ablation) generated from mouse embryos[22,24]. Human squamous cell carcinoma SCC4, 9, and 15 cells were cultured in 10% FBS/DMEM-F12. All the cells were grown under 37 °C with 5% CO$_2$ condition. Where indicated, primary lung cells were isolated from T and B cell-deficient NOD-SCID mice and cultured using 10% FBS/RPMI. These cells were WWOX-positive and used for the migration assay as described below. Supporting evidence revealed that human cancer MDA-MB-435s cells appears to be derived from melanoma rather than from breast cancer[58]. Mouse experiments were carried out as previously described[16]. Organs were harvested, stored in a −80 °C freezer, and used for last experiments[16] and indicated experiments in this study.

**Chemicals and antibodies.** Chemicals used in this study were: (1) Propidium iodide, sodium orthovanadate, hydrogen peroxide, and proteinase inhibitor cocktail from Sigma Aldrich (St. Louis), (2) DAPI (4′,6-diamidino-2-phenylindole) and TGF-β RI Kinase Inhibitor II from Merck Millipore (Burlington, MA), (3) Green fluorescent calcium binding dye Fluo-8 for live cell imaging from AAT Bioquest (Sunnyvale, CA), (4) ceritinib (LC Labs, Boston), (5) estrogen receptor agonist CI-4AS-1 from Santa Cruz Biotechnology (Dallas, TX), and (6) RedoxSensor Red CC-1 stain from Molecular Probes/Invitrogen (Carlsbad, CA). Recombinant TGF-β1 and TGF-β2 were from Preprotech. Polyclonal antibodies against GFP and monoclonal antibodies against TβRII and Flotillin-2 were from Santa Cruz Biotechnology. Monoclonal antibody against WWOX was from Abnova and Santa Cruz Biotechnology. The following polyclonal antibodies were generated in rabbits using the following synthetic peptides (Genemed Synthesis)[37]: (1) gre (WWOX7-21): CAGLDDTDSEDELPPG; (2) pS14-gre (pS14-WWOX7-21): CAGLDDTDp-SEDELPPG; (3) repl (WWOX286-299): DYWAMLAYNRSKLC; (4) pY287-repl (pY287-WWOX286-299): DpYWAMLAYNRSKLC. The first cysteine in the gre and pS14-gre peptides was added for covalent coupling with KLH (Keyhole limpet hemocyanin)[34]. Validation of the produced antibodies was tested in Western blotting, peptide blocking in immunohistochemistry, and co-immunoprecipitation.

***Wwox* cDNA constructs.** We reported the construction of the full-length murine Wwox cDNA in pEGFP-C1 (Clontech) and cDNAs coding for the N-terminal first and second WW domains (WW1 and WW2) and the C-terminal SDR domain, respectively, in pEGFP-C1 or other vectors[3,34]. Additional Wwox constructs were made, including (1) WWOX(1-21)-pEGFP-C1 Forward, 5′-TCGAATTCTATGG CAGCTCTGCGCTATGCG and Reverse, 5′-TGGAATTCCTAGCCCGGAGGC AGCTCATC; (2) WWOX(7-21)-pEGFP-C1, Forward, For: CTGAATTCAGCGG GCCTGGACGACAC, Rev: TGGAATTCCTAGCCCGGA GGCAGCTCATC. Reverse transcription and polymerase chain reaction (RT/PCR)[3,34] was carried out with an annealing temperature at 62 °C. The restriction site was EcoR1 in the PCR product for ligating into pEGFP-C1, as described[3,34].

**Western blotting and immunofluorescence microscopy.** Standard Western blotting analyses were carried out, as described[7,15,16,25]. In brief, protein sample preparations were quantified (BCA protein assay; ThermoFisher Scientific) and 40 μg of each sample was loaded onto SDS-polyacrylamide gels, followed by electrophoresis under reducing conditions. Following electroblotting separated proteins onto nitrocellulose membranes (BioRad), the membranes were probed with a specific mouse or rabbit antibody, followed by washing and probed with a secondary antibody conjugated with horseradish peroxidase (ThermoFisher Scientific). Where indicated, immunofluorescence microscopy using specific primary mouse or rabbit antibodies was carried out[7,15,16,25]. Secondary antibody against the

primary mouse or rabbit antibodies was conjugated with red fluorescent rhodamine, Texas red, or Alexa 555 (Molecular Probes/Invitrogen or Jackson Laboratory). Alternatively, the green fluorescent secondary antibody was conjugated with fluorescein or Alexa Fluor 488 (Molecular Probes/Invitrogen or Jackson Laboratory). Appropriate secondary antibody combinations were matched for dual antibody staining to avoid cross-react[15–17].

**Cell migration assay and time-lapse microscopy**. Cell migration assay was performed using culture-inserts (ibidi, Verona, WI)[38]. Each insert has two chambers, with a separating distance of $500 \pm 50\,\mu m$, for seeding cells (70 μl, $4 \times 10^5$ cells each side) on 35-mm Petri dishes. Indicated cells were seeded in each chamber and cultured overnight at 37 °C/5% CO2. The insert was then gently removed with a pair of forceps, and the medium replaced with serum-free medium or containing 2% FBS. Where indicated, 10% FBS or serum-free was used. Time-related cell migration distance and velocity were measured using a NIKON TE2000-U microscope or an Olympus IX81 microscope with autofocus[38]. Ten to twenty randomly selected cells were measured to determine the mean velocity and accumulative migration distance. Single cell moving path was tracked by using the tracking, chemotaxis, and migration tools of Image J (NIH, Bethesda, MD).

**Bubbling cell death assay by time-lapse microscopy**. Live WWOXf or WWOXd cells were added an aliquot of Fluo-8 (50 μM) and nontoxic levels of PI (2 μg/ml) and DAPI (10 μg/ml). Cells were exposed to UV irradiation (240–960 mJoule/cm$^2$). Time-lapse microscopy at 200x magnification was then carried out to image calcium influx and BCD[39]. One picture was taken per 2 min. Similarly, WWOXd cells were treated similarly for time-lapse microscopy.

**Velocity autocorrelation function (VACF)**. Where indicated, VACF was calculated to confirm the cell migration and the effect of peptides[38,54]. The formula of calculation was:

The velocity is expressed as $\vec{v}\,(t) = [\vec{R}\,(t + \delta) - \vec{R}\,(t)]\,\delta$

The velocity autocorrelation function is defined as $R(\tau) = \langle \vec{v}\,(t + \tau)\cdot\vec{v}\,(t)\rangle$

Where, $\vec{R}\,(t)$ means the position of a single cell tracking center. τ means the time interval of the normal diffusion. δ means the time interval of each frame. The maximum correlation value is 1. The lower the value means more uncorrelated. We used the temporal correlation of the velocity to randomly analyze the behavior of each cell migration using 10 cells. VACF is calculated by the aforementioned formula[38,54,59].

**Förster (Fluorescence) resonance energy transfer (FRET)**. FRET analysis was performed as described previously[7,32,37,59]. The following mammalian expression constructs were made: (1) the N-terminal WW (1st and 2nd) domains in pEGFP-C1 and pDsRed2-monomer, and (2) the C-terminal SDR domain in pEGFP-C1 and pDsRed2-monomer (Clontech). COS7 cells were transfected with both constructs by liposome-based Genefector (VennNova) and cultured for 24–48 h. FRET analysis was performed using an inverted fluorescence microscope (Nikon Eclipse TE-2000U), with an excitation wavelength 440 nm for donor EGFP and an emission wavelength 535 nm for acceptor DsRed monomer for measuring the FRET signal. The FRET images were corrected for background fluorescence from an area free of cells and spectral bleed-through. The spectrally corrected FRET concentration (FRETc) was calculated by Youvan's equation (using a software program Image-Pro 6.1, Media Cybernetics): FRETc = [fret − bk(fret)] − cf(don) × [don − bk(don)] − cf(acc) × [acc − bk(acc)], where fret = fret image, bk = background, cf = correction factor, don = donor image, and acc = acceptor image. The equation normalizes the FRET signals to the expression levels of the fluorescent proteins. Where indicated, determining 3-protein binding during signaling was carried out by time-lapse FRET analysis[37,59]. For example, MEF Wwox knockout cells were transiently overexpressed with ECFP-IκBα, EGFP-ERK and DsRed-WWOX using liposome-based GeneFector (VennNova), followed by culturing for 24–48 hr. Wild type MEF cells in suspension were added to the knockout cell monolayers to induce the IκBα/ERK/WWOX signaling in the knockout cells, as measured by time-lapse microscopy[37,59]. The energy flow goes from ECFP, EGFP and down to DsRed. In negative controls, non-tagged ECFP, EGFP and DsRed constructs were used.

**Cell proliferation assay**. A total of $5 \times 10^4$ cells were seeded in a 12-well microtiter plate. Cells were cultured in medium containing 10% FBS overnight. Cells were then trypsinized and stained with trypan blue, and the cell number was counted using a hemocytometer at time point 0, 6, 18, 24, and 48 hr, respectively.

**Peptide coating for ELISA**. A 96-well microtiter plate was coated with 10 μg bovine serum albumin (BSA) in 35 μl coating buffer (3.7 g Sodium Bicarbonate, 0.64 g Sodium Carbonate in 1 L distilled water), or 35 μl of 1:100 diluted heat-inactivated FBS, and incubated at 37 °C for 1 hr, followed by washing twice with PBS. Peptides (15 ng in 40 μl MilliQ or ~400 nM for a 13-amino-acid peptide) were added to each well of the microtiter plate and incubated at 37 °C for 16 hr, followed by washing and blocking with 5% BSA in PBS at room temperature for an hour. The plate was washed three times with PBS-T (PBS contain 0.05% Tween-20), followed by adding primary antibody (1:300 in PBS-T) and incubating at room

temperature for 2 hr. The plate was then washed with PBS-T for 6 times, then added secondary antibody (1:5000 in PBS-T) and incubated at room temperature for 1 hr, followed by washing with PBS-T and adding 50 μl 3,3′,5,5′-Tetramethylbenzidine (TMB) substrate. The reaction was stopped by adding 50 μl HCl.

**Animal use**. All experiments involved in mouse use have been approved by the Institutional Animal Care and Use Committee (IACUC) of the National Cheng Kung University College of Medicine (Approval numbers 105064, 105070, 106064, 107027, 107080, 107296, 108041, 108153, and 110001).

**Statistics and reproducibility**. Data were analyzed by one-way Anova and Student's t test (two-tail analysis with equal or unequal variances) using Microsoft Excel and Prism 7. Data were expressed as mean ± standard deviation. All experiments were repeated at least 3 times. Each experiment may contain 10–20 independent data points. $p < 0.05$ was considered as statistically significant.

**Reporting summary**. Further information on research design is available in the Nature Research Reporting Summary linked to this article.

## Data availability

The authors declare that all the data from our experiments supporting the findings of this study are available within this article and its supplementary information files. Also, data are available upon reasonable requests to the authors. Spreadsheets and Supplementary Videos 1–12 and Supplementary Videos 13–42 for Supplementary Figs. 2 and 5 are deposited at Figshare (https://doi.org/10.6084/m9.figshare.14560782). Legends for all videos is shown in the Description of Additional Supplementary Files.

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

## Acknowledgements

This research was supported to NS Chang by the Ministry of Science and Technology, Taiwan (MOST 105–2320-B-006-046, 105–2320-B-006-036, 106–2320-B-006-061, 106–2320-B-006-017, 107-2320-B-006-058-MY3, and 107-2320-B-006-005) and the National Health Research Institute (NHRI-EX107-10734NI). Yong-Da Sie was the recipient of the Graduate/Postdoctoral Travel Award from the American Society for Biochemistry and Molecular Biology (ASBMB) in April 2017. Yu-An Chen was supported by a grant from the Taiwan Foundation For Rare Disorders. Tsung-Yun Liu was the recipient of the Young Investigator Award from the American Society of Pharmacology and Experimental Therapeutics (ASPET) in April 2020.

## Author contributions

Y.A.C., Y.D.S., T.Y.L., H.L.K., Y.J.C., and P.Y.C.: Migration assays; Y.A.C., K.T.L., and P.J.C.: Western blotting and immunohistochemistry; H.L.K.: FRET analysis; S.T.C.: Immunoelectron microscopy; N.S.C., P.J.C., Y.J.C., and T.Y.S.: Co-immunoprecipitation; Y.A.C. and T.Y.L.: Protein coating and ELISA assay; N.S.C. and T.Y.S.: BCD and explosion assays; N.S.C.: Conceived ideas, coordinated teamwork, carried out experiments and imaging analyses, and wrote the manuscript. Specifically, Fig. 1a–l contributed by T.Y.L., Y.J.C., Y.D.S., and 1m–p by NSC; Fig. 2 by H.L.K.; Fig. 3a–f by T.Y.L., Y.J.C., Y.D.S., and 3g by Y.A.C.; Fig. 4a–b by N.S.C. and 4c–e by Y.J.C., P.Y.C., T.Y.L., N.S.C.; Fig. 5 by Y.A.C., Y.D.S., P.Y.C.; Fig. 6 by Y.A.C.; Fig. 7 by Y.A.C.; Fig. 8 by Y.A.C., Y.D.S.; Fig. 9 by Y.A.C.; Fig. 10a–e by P.Y.C. and 10f–i by Y.A.C., N.S.C.; Supplementary Fig. 1 by Y.A.C.;

Supplementary Fig. 2 by N.S.C.; Supplementary Fig. 3a by P.Y.C., Y.A.C., Y.D.S., T.Y.L., Y.J.C., and 3b–c by K.T.L.; Supplementary Fig. 4 by Y.J.C.; Supplementary Fig. 5a by N.S. C., 5b by K.T.L., 5c–f by T.Y.L., Y.J.C.; Supplementary Fig. 6 by H.L.K.; Supplementary Fig. 7 by T.Y.L., Y.J.C., Y.D.S.; Supplementary Fig. 8 by T.Y.L., Y.J.C.; Supplementary Fig. 9a–d by T.Y.L. and 9e by Y.J.C., T.Y.L.; Supplementary Fig. 10–13 by Y.A.C.; Supplementary Videos 1–3 by T.Y.L., Y.J.C., Y.A.C., Y.D.S., P.Y.C.; Supplementary Videos 4 and 5 by N.S.C.; Supplementary Videos 6 and 7 by T.Y.L., Y.J.C., Y.A.C., P.Y.C.; Supplementary Videos 8–12 by H.L.K.; Supplementary Videos 13–42 by N.S.C.

## Competing interests
The authors declare no competing interests.
