## [Transparent Peer Review File · Communications Biology]

Reviewers' comments:

Reviewer #1 (Remarks to the Author):

Recommendation: This research represents an interesting and potentially important series of experiments investigating the roles of different epitopes of the tumor suppressor Wwox gene in cell recognition, and how TGF β is involved in this process. Most of the data support the conclusion. But some specific mechanisms still need further research and there are some minor issues that need correction.

Specific Comments:

- When verifying the cell migration experiment in this manuscript, the serum concentration of the culture medium is 2% to inhibit the proliferation and division of cells. In the process of tumor development, cancer cells proliferate indefinitely. Does the phenomenon of Wwox-positive cells repelling invading Wwox-negative cancer cells still exist?
- The anchoring of Wwox on the membrane can be achieved by combining with Hyal-2 or self-polymerization. How does these two anchoring modes affect Wwox to perform different functions? Please verify, which would increase the clarity and significance their results?
- To determine the effect of TGF β signaling on repelling Wwox-negative cells, Wwox negative and Wwox positive + T β Rii IgG were used for comparison. Is this rigorous? I suggest that Wwox negative, Wwox positive, and Wwox positive + T β Rii IgG should be used to illustrate this conclusion.
- Wwox 7-21 peptide promote the greeting and migration of cells. Meanwhile, the phosphorylation of S14 could inhibit this promotion either. The result shows us that S14 plays an important role in the process of cell greeting and migration. However, the 11-20 peptide has no effect on cell migration. Why?
- In Figure 1h, the peptide was tested for toxicity to cell growth using 2% FBS. While 2% FBS only guarantees the survival of cells and inhibits their division, In this regard it is unclear to me what is the significance of detecting the cell cycle under different peptide treatments under 2% FBS?
- In Figure 2, the coating efficiency of peptides was detected by ELISA. The four peptides (gre, pS14gre, repl, pY287repl) were not completely consistent. Should the influence of the four peptides on cell migration be tested under the same coating efficiency?
- In Figure 3c, the gre signal should be localized in the cell membrane, but appears in the nucleus in the merge image. Please specify
- What effect does the SDR domain have on the migration of the cell itself? From the comparison of Figures 4b and 4c, compared with EGFP, cells expressing SDR basically do not migrate.
- Please explain why there seems to be no statistical differences in Figure s2, s3,1c,2d and 7g.
- In the schematic diagram shown in Figure 8d, phosphorylation at site 33 of Wwox leads to the exposure of the repl peptide, but it seems that there are no same issues in the manuscript to support this conclusion.
- What role does the 287 site of Wwox play in cell migration? On line 183 of the manuscript, phosphorylation of Y287 can reduce cell migration by nearly 50%. Whereas on line 201 of the manuscript, Y287 phosphorylation is not needed for cell repulsion. Why?

Reviewer #2 (Remarks to the Author):

Comments to Manuscript Chen et al.

In the present study Chen et al. investigated the role of the surface epitope WWOX286-299 and concluded from their results that this epitope is responsible for repelling WWOX-negative cancer cells. In brief, this is an interesting study. However, some parts of the manuscript are not really consistent and it seems that as much data as possible were summarized here. Different cell lines were compared

with each other making it difficult to conclude that WWOX286-299 is truly responsible for repelling. Likewise, too much previous work has been cited in the results section, which rather belongs to the Introduction or Discussion section. I am very sorry, but the manuscript is not yet suitable for publication in its present form.

Major concerns:

1) A major concern is that only one WWOX negative cell line was used for the experiments. At least one additional WWOX negative cancer cell line should be used to substantiate their findings.

2) It remains unclear why the authors have presented data summarized in Fig. 1. Here, data for SCC9, SCC15 cells, MEF cells and HCT116 cells are shown, whereas the MDA-MB-231 breast cancer cell line was used for most of the following experiments. Please explain.

In this context it remains ambiguous why SCC9 cancer cells were defined as WWOX negative even though WB data in ref#38 clearly showed that a weak WWOX expression was detected in SCC9 cells. Maybe the authors could also comment on this.

Furthermore, it remains unclear why Hyal-2 and WWOX immunoelectron microscopy data of HCT116 were shown here because this has already been done in ref#7. Please explain.

Likewise, line 89-91: "WWOX-deficient human MDA-MB-231 breast cancer cells migrated backward upon facing WWOX-positive mouse L929 fibroblasts (Suppl Video 1)". Appropriate data have been already shown in Ref #37 (Fig. 5a: MDA-MB-231 underwent retrograde migration upon encountering L929). Please explain.

3) Even though it seems to be clear that MDA-MB-231 breast cancer cells lack WWOX expression appropriate Western Blot data should be presented. Likewise, appropriate TbetaRII and Hyal2 WB data should be presented since TGF-beta binds to both.

4) The authors should show present appropriate migration data for MDA-MB-231 cells with TbetaRII IgG. Otherwise, it cannot be concluded that MDA-MB-231 cells behave similar to MEF-/- cells.

5) The "Abstract" has to be rewritten since too much previous work is summarized here (e.g., line 22-24: "To compromise, WWOX- cells secrete" and line 26-27: "In contrast, when WWOX+ cells were treated ..."). These passages are rather related to the findings of ref#37, but not to the data presented here.

6) Fig. 4a: The authors should describe more in detail what can be seen on the Western Blot. As written in line 184-185 "stable transfectants of MDA-MB-231 cells expressing EGFP-tagged SDR domain or EGFP only" were established. However, it seems that additional EGFP-tagged proteins were established, such as EGFP-WWOX1-21, EGFP-WWOX7-21 and EGFP-WW domains. Please explain. In this context it remains unknown which housekeeping gene has been used for control purposes.

7) For me it is difficult to conclude that Flotillin is colocalized with SDR domain in lipid rafts because the resolution of the images is too low and the brightness of the green and red fluorescence is too high. In this context, it remains unclear why the green and red intensity of Fig. 4b is much higher than of Suppl. 5a and 5e. Conceivably, this might be attributed to the pdf conversion process.

8) The authors should explain more in detail why MEF cells have been used for data summarized in Fig. 7 and not MDA-MB-231 cells. Likewise, it remains unclear why additional experiments with TGF-beta2 have been done. This should be briefly explained. Likewise, appropriate experiments with TGF-beta2 and MDA-MB-231 cells should be performed. Moreover, it would be great if the authors could briefly discuss why TGF-beta2 suppressed the migration of knockout MEF cells (Fig. 7b, c). This is missing in the discussion.

9) What is the effect of TGF-beta2 on the proliferation of the MEF wildtype and knockout cells?

10) Statistical analysis: I strongly recommend to the authors to recalculate the statistics using an appropriate method (either one-way ANOVA (if data are Gaussian distributed) or a nonparametric test (e.g., Kruskal-Wallis) with an appropriate post-hoc test).

Minor comments:

1) line 89-91: "WWOX-deficient human MDA-MB-231 breast cancer cells migrated backward upon facing WWOX-positive mouse L929 fibroblasts (Suppl Video 1)". Appropriate data have been already shown in ref #37 (Fig. 5a: MDA-MB-231 underwent retrograde migration upon encountering L929).

2) Error bars are missing in Fig. 1c, Fig. 2d, Fig. 4i, and Fig. 7g. Please correct.

3) Mat & Method section: cell culture conditions for SCC-9 and SCC-15 cells are missing.

4) Line 410: "liver and lung cells were": no experiments with liver cells were performed here.

5) Instead of referring to previous published work a brief description for Western Blot experiments and immunofluorescence microscopy including the used antibodies and fluorochromes should be given here.

6) Line 293: "MDA-MB-231 cells to form cancer stem cell-like spheres". It is commonly known that MDA-MB-231 breast cancer cells rather form cell clumps than spheres (Manuel Iglesias et al. 2013, PLoS ONE 8:e77281).

January 22, 2021

Editor
Communications Biology

Dear Editor:

Thank you for kindly managing our manuscript. We appreciate the outstanding efforts contributed by both reviewers. At the request of the reviewers, we have thoroughly revised our manuscript. We have indeed addressed each comment in great details. That is, many new discoveries have come up as a result of reviewers' great questions and requests.

Reviewer #1 (Remarks to the Author):

Recommendation: This research represents an interesting and potentially important series of experiments investigating the roles of different epitopes of the tumor suppressor Wwox gene in cell recognition, and how TGF β is involved in this process. Most of the data support the conclusion. But some specific mechanisms still need further research and there are some minor issues that need correction.

Answer: Thank you for your outstanding efforts. As requested, this manuscript has been thoroughly revised. Also, as mentioned in the beginning, we have added new data and established advanced concepts regarding the physiological features of functional WWOX-expressing (WWOXf) cells versus WWOX-dysfunctional or -deficient (WWOXd) cells. We have determined the roles of the functional WWOX in calcium influx, bubbling cell death, and cell migratory behavior:

WWOXf cells possess functional WWOX and are able to carrying out calcium influx, bubbling cell death at room temperature, and cell migration repulsion for WWOXd cells.

WWOXd cells are capable of undergoing explosion in response to stress stimuli at room temperature (independently of calcium influx), may migrate backward upon facing WWOXf cells, kill WWOXf cells without physical contacts, and utilize ERK/WWOX/I κ B α signaling for survival.

Specific Comments:

1. *When verifying the cell migration experiment in this manuscript, the serum concentration of the culture medium is 2% to inhibit the proliferation and division of cells. In the process of tumor development, cancer cells proliferate indefinitely. Does the phenomenon of Wwox-positive cells repelling invading Wwox-negative cancer cells still exist?*

Answer: As requested, we have increased the fetal bovine serum concentration up to 10%, as in the standard cell culture media. Our data showed that TNF-resistant WWOXd L929R cells migrate reluctantly and some cells undergo retrograde migration upon facing WWOXf L929S cells using 10% FBS (new Fig. 1). This observation is similar to the culture conditions using 0 and 2% FBS

(new Fig. 1). Indeed, the higher the serum concentration, the slower the migration of L929S cells (new Fig. 1). L929R cells induced apoptosis of L929S cells from a distance (~50% death). The death effect is in agreement with our previous observations using other pairs of WWOXf versus WWOXd cells (ref. 38). Similar results were observed by testing MEF wild type versus *Wwox* knockout cells at 10% FBS (Supplementary Fig. 1g and Video S1). That is, the *Wwox* knockout MEF cells underwent retrograde migration upon facing the wild type MEF cells.

Please note that there are additional cell-pair data summarized in the revised Table I (original version). At the request of Reviewer 2, we have added a panel of new cell line pairs (18 pairs; new Supplementary Fig. 3a). WWOX protein expression is also shown (Supplementary Fig. 3b,c). Unless otherwise indicated, all cell migration conditions were set using 2% FBS/medium.

Observations from few randomly selected cell pairs during migration are summarized as follows:

a) L929S fibroblasts versus L929R fibroblasts. L929S is sensitive to TNF-killing, and L929R resistant (*J. Biol. Chem.* 270,7765-7772, 1995). Expression of WWOX in L929S is low, but is inducible by hyaluronidase, TNF, UV and other stress stimuli (*J Biol Chem.* 2001;276:3361-70; *J Biol Chem.* 2003;278:9195-202; *J Biol Chem.* 2005;280:43100-8) (Supplementary Fig. 3b).

b) L929S versus MDA-MB-231. We have used this cell pair for many experiments, as shown in the original and this revised manuscript. Breast WWOXd MDA-MB-231 cells express low levels of WWOX (Supplementary Fig. 3b,c), do not possess an efficient system for calcium influx (Supplementary Fig. 2d), and undergo explosion in response to stress stimuli such as ceritinib and UV/cold shock (*Cancers (Basel).* 2019;11:1818).

c) Squamous cell carcinoma WWOXf SCC4 or 9 versus WWOXf SCC15 (Tsai et al., *Cell Death & Disease* 2013;4:792) (Supplementary Fig. 3a13, 5a-f).

d) WWOXd breast MDA-MB-435s cells versus wild type MEF cells (Supplementary Fig. 3a14).

e) WWOXd 13-06-MG or U87-MG glioblastoma cells versus L929S cells (Supplementary Fig. 3a10,11).

2. The anchoring of Wwox on the membrane can be achieved by combining with Hyal-2 or self-polymerization. How does these two anchoring modes affect Wwox to perform different functions? Please verify, which would increase the clarity and significance their results?

Answer: Thank you for this outstanding point. Without three dimensional information from each protein and their binding interactions, it is impossible to give a clear answer.

Please note that WWOX binds Hyal-2, Ezrin (*Biochem Biophys Res Commun.* 2006;341:784-91) and Merlin/NF2 (unpublished) on the cell surface. Also, WWOX can undergo self-polymerization (ref. 7,32). These observations have been validated by immunoelectron microscopy (ref. 7,32). By FRET microscopy, we determined the binding of WW domain with SDR domain and self-binding WW domain and SDR domain (new Fig. 10f). How the three-dimensional structure of WWOX looks like remains to be established yet.

To answer your question, we have carried out co-immunoprecipitation using organs, and showed the presence of WWOX, Hyal-2, T β RII, and Flotillin-2 complex in the precipitates (Fig. 4c-e), suggesting that the WWOX/Hyal-2/T β RII signaling complex is in the lipid raft. This complex is large enough to be visualized by cryo-electron microscopy. We plan to work on this and publish

our data later on. We have also used cancer cell lines to run co-immunoprecipitation. However, due to variations in protein expression and potential modifications among cell lines. It is difficult to jump into a conclusion.

By yeast two-hybrid analysis, we have shown that membrane Hyal-2 binds the *N*-terminal WW domains of WWOX in a Tyr33 phosphorylation-dependent manner (ref. 32). That is, binding of Hyal-2 with pY33-WWOX is essential to initiate signaling. In our recent report (ref. 38), we determined that Hyal-2 antibody induces anterograde migration of MEF *Wwox* knockout toward wild type cells, and thereby apoptosis of wild type MEF cells caused by knockout MEF cells is abolished (ref. 38).

Supporting data using specific antibodies against pY33-WWOX and TβRII are stated below. (see Answer #3). Together, TβRII IgG retards the occurrence of BCD and extends the duration of cell death.

3. *To determine the effect of TGFβ signaling on repelling Wwox-negative cells, Wwox negative and Wwox positive + TβRii IgG were used for comparison. Is this rigorous? I suggest that Wwox negative, Wwox positive, and Wwox positive + TβRii IgG should be used to illustrate this conclusion.*

Answer: As requested, we have carried out the following experiments using 3 cell pairs, which are MEF wild type versus *Wwox*^{-/-} knockout cells, L929R versus L929S, and MDA-MB-231 versus L929S.

- 1) During cell migration for 48 to 72 hr, both MEF wild type and *Wwox*^{-/-} knockout cells were exposed to antibodies against TβRII as measured by time-lapse microscopy.
- 2) The MEF *Wwox*^{-/-} knockout cells were pretreated with TβRII IgG antibody for 24 hr, followed by washing and processing migrating assay versus the wild type cells. That is, both time-lapse microscopy and end-point determination of cell migration and apoptosis in 48-72 hr were carried out.
- 3) Under similar conditions, TβRII IgG antibody was used to treat the wild type MEF cells for 24 hr followed by processing time-lapse microscopy or end-point migration assay.

Our data are shown in the new Fig. 3 and Supplemental Fig. 8. In short, when WWOXf (e.g. L929S or MEF wild type) cells were pretreated with TβRII IgG or pY33-WWOX antiserum, these cells induce the anterograde migration of WWOXd cells (e.g. L929R or MEF *Wwox* knockout). Both cell pairs became merged and invaded each other's area (Fig. 3c). No apoptosis occurred. Without antibody, WWOXd cells caused apoptosis of WWOXf cells (Fig. 1d,h,i; 5b; Supplementary Fig. 8b-d). When L929 or MEF cell pairs were cotreated with TβRII IgG or pY33-WWOX antiserum during cell migration, both cell pairs survived and invaded each other's area (Fig. 3d,g; Supplementary Fig. 8b-d). Intriguingly, when L929R or MEF *Wwox* knockout cells were pretreated with TβRII IgG or pY33-WWOX antiserum, apoptosis occurred mainly in the L929R or MEF *Wwox* knockout cells (Fig. 3f,g; Supplementary Fig. 8b-d). Normal rabbit serum controls were used (Supplementary Fig. 8d).

4. *Wwox 7-21 peptide promote the greeting and migration of cells. Meanwhile, the phosphorylation of S14 could inhibit this promotion either. The result shows us that S14 plays an*

important role in the process of cell greeting and migration. However, the 11-20 peptide has no effect on cell migration. Why?

Answer: In our recent report (ref. 33), we determined that synthetic WWOX7-21 peptide (AGLDDTSEDELPPG), or truncated to 5-amino acid WWOX7-11 (AGLDD), significantly suppresses and prevents the growth and metastasis of melanoma and breast and skin cancer cells in mice. Compared to WWOX7-21 peptide, pS14-WWOX7-21 peptide has lost its activity in enhancing cell migration. Although during the initial stage of cell migration (less than 18 hr), pS14-WWOX7-21 peptide may slow down the cell migration (new Fig. 5f). After that, pS14-WWOX7-21 peptide had no significant effect in suppressing cell migration, compared to controls (Fig. 5f). Please note that in contrast to WWOX7-21 peptide-mediated growth suppression, pS14-WWOX7-21 peptide strongly accelerates cancer cell growth (ref. 33).

The sequence of WWOX7-11 could represent one of the functional motifs in WWOX to exert cancer suppression. Here, we showed that WWOX7-21 peptide enhances cell migration and provides a greeting signal for cell-to-cell recognition. Without the WWOX7-11 motif, WWOX11-20 peptide is not functional in enhancing cell migration (new Fig. 5f).

In a new set of experiments, we treated MEF wild type or *Wwox* knockout cells with the WWOX7-11 peptide and showed its efficacy in enhancing cell-to-cell greeting and blocking knockout cell-mediated apoptosis of MEF wild type cells (new Figure 5b-e). The observation is similar to that of the effect of gre peptide (WWOX7-21) (new Figure 5f,g,i). Additional data showed that WWOX7-11 is most potent in enhancing U87-MG cell migration (new Supplementary Fig. 9d). Again, “AGLDD” in WWOX7-11 is a motif in WWOX for enhancing cell migration and cell-to-cell recognition, as well as cancer suppression (ref. 33). How WWOX7-11 works to coordinate cell migration and recognition and cancer suppression remains to be established.

5. *In Figure 2h, the peptide was tested for toxicity to cell growth using 2% FBS. While 2% FBS only guarantees the survival of cells and inhibits their division, In this regard it is unclear to me what is the significance of detecting the cell cycle under different peptide treatments under 2% FBS?*

Answer: Thank you. It is a good point. As requested, we have decided to delete this data. We have already tested the anticancer functions for all the indicated peptides *in vivo*, except for WWOX11-20 and WWOX2(353-363) (Cancers (Basel). 2019;11:1818; Cancers (Basel). 2020;12:E2189). None of the peptides are toxic to mice. Our unpublished data showed that inhibition of breast cancer cell growth *in vivo* by WWOX7-21 and WWOX7-11 peptides is due to peptide-mediated activation of memory cytotoxic Z lymphocytes. Z cells are not T, B, NK cells, or macrophages (ref. 15-17).

6. *In Figure 2, the coating efficiency of peptides was detected by ELISA. The four peptides (gre, pS14gre, repl, pY287repl) were not completely consistent. Should the influence of the four peptides on cell migration be tested under the same coating efficiency?*

Answer: We agree with your point. Please note that short peptides of 12~15 amino acids in length with various compositions cannot exhibit stable tertiary structures in solution. Coating of these

peptides to plastic surface or serum-precoated matrix cannot guarantee that each peptide can adhere to the matrix in a similar fashion. Indeed, it is difficult to reach an identical amount of each peptide coated onto the matrix. The original Figure 2c, dealing with peptides coated onto serum-precoated plastic surface is now shown in Supplementary Fig. 9a-c. Various experimental conditions have been carried out.

As requested, in new coating experiments, we used U87-MG cells and 10 WWOX peptides. Peptide coating efficiency was similar among all peptides, except that pY287-WWOX286-299 had the best coating efficiency. Compared with other synthetic WWOX peptides, WWOX7-11 and WWOX7-11A7R strongly enhanced U87-MG cell anterograde migration to each indicated peptide, and that WWOX7-11G8R lost the enhancing activity (new Supplementary Fig. 9d). The observations suggest a critical role of Gly8 in supporting the enhancing activity of the gre (or WWOX7-21) epitope. WWOX286-299 (or repl) provided the best repellence activity for U87-MG (new Supplementary Fig. 9d,e) and MDA-MB-231 (new Fig. 5f). pY287-WWOX286-299 lost its repellence activity. Reducing the amount of pY287-WWOX286-299 in coating still showed the loss of repellence activity (data not shown). Overall, our data suggest that phosphorylation of Y287 in repl of WWOX significantly increased its binding to the plastic surface coated with or without albumin or serum proteins, and reduces repl's repellence activity.

7. In Figure 3c, the gre signal should be localized in the cell membrane, but appears in the nucleus in the merge image. Please specify.

Answer: Please note the original Fig. 3c is now Fig. 6c. Please note that WWOX protein is present in many subcellular organelles in cells, including cytoplasm, nucleus, lysosome, mitochondrion, and cell membrane (Ref. 3,4,7,8,9,13). Based on the study from us and others since 2000, our group has first demonstrated that WWOX protein can be present in the cytoplasm and nucleus (Ref. 3). That is, under stress conditions (e.g. UV irradiation, chemotherapeutic drugs and TNF), WWOX relocates, together with p53, to the mitochondria and nuclei to induce apoptosis (Ref. 3,4,34,36). Under physiological conditions, TRAPPC6A acts as a carrier for WWOX to enter the nucleus (Ref. 25,26). There is a nuclear localization signal "KRKR" in between the N-terminal first and second WW domains of WWOX.

WWOX does not have a membrane localization signal. Membrane WWOX is believed to be anchored by membrane/cytoskeletal proteins such as Hyal-2, Ezrin (Ref. 4) and Merlin/NF-2 (unpublished). Presumably, self-polymerization of WWOX allows the stacking of WW domain area, which possesses β -sheets to localize in the membrane lipid-rich microenvironment. Taken together, the gre signal is in front of the first WW domain. Transiently overexpressed WW domains plus the N-terminal leader sequence (including gre) allows nuclear localization (Ref. 3,34). Again, how WWOX localizes in the membrane is a mystery to solve.

8. What effect does the SDR domain have on the migration of the cell itself? From the comparison of Figures 4b and 4c, compared with EGFP, cells expressing SDR basically do not migrate.

Answer: We agree with your point. EGFP-SDR-expressing cells are much less mobile (new Fig. 7d). And, this reduced cell migration is reversed by serum factors, so that cells can migrate individually (new Fig. 7e). Please also refer to data in new Figure 1a,e,I, which shows serum

factors limits the migration of L929S cells. SDR antibody did the similar effect in enhancing cell migration (new Fig. 7f). Our supporting data showed that SDR domain binds another SDR domain or WW domain, and WW domain interacts with another WW domain (new Fig. 10f). Presumably, this may result in the formation of a net-like network on the cell surface. This network interacts with integrins, makes the cells less mobile, and renders them to undergo collective migration. Indeed, by gene chip analysis, we found that *Wwox* gene knockout MEF cells have significant reduction in adhesion molecules and integrins (unpublished).

From the Repl peptide experiments (new Supplementary Fig. 9), when Repl is phosphorylated at Y287, there is an increased binding of the peptide to the extracellular matrix. Conceivably, the full-length protein pY287-WWOX has an increased affinity to extracellular matrix, so as to limit cell migration. We have added this point in the Discussion (page 21).

9. *Please explain why there seems to be no statistical differences in Figure s2, s3, 1c, 2d and 7g.*

Answer: As requested, we have added the standard deviations in each figure and calculated the statistical differences by one-way Anova. Please note Supplementary Figure s2 is now deleted; Fig. s3 is s1f; Fig. 1c deleted and replaced with Fig. 3b-e; Fig. 2d is 5g, and Fig. 7g deleted. Those deleted are not necessary in this revised manuscript.

10. *In the schematic diagram shown in Figure 8d, phosphorylation at site 33 of Wwox leads to the exposure of the repl peptide, but it seems that there are no same issues in the manuscript to support this conclusion.*

Answer: We have decided to delete this figure. Molecular tools will be needed to elucidate the structure and functional relationship. Currently, the tertiary structure of WWOX is not known.

11. *What role does the 287 site of Wwox play in cell migration? On line 183 of the manuscript, phosphorylation of Y287 can reduce cell migration by nearly 50%. Whereas on line 201 of the manuscript, Y287 phosphorylation is not needed for cell repellence. Why?*

Answer: Sorry for the confusing. From our data, it is quite clear that repl peptide (WWOX286-299), which is in the SDR domain of WWOX, contributes to repellence to WWOXd cells. However, when Y287 is phosphorylated, the repellence activity of repl is reduced (new Fig. 5f). The functional reduction in repellence is due to Y287 phosphorylation. After phosphorylation, pY287-WWOX is readily subjected to ubiquitination and degradation (Ref. 44). Based upon our peptide coating experiments, pY287-WWOX protein has an increased capability in adhesion and this may limit cell migration. Please note that the original Line 201 has been reworded as “When S14 is phosphorylated, pS14gre (pS14-WWOX 7-21) peptide had lost its enhancing activity of cell migration as compared to controls (Fig. 5f).” (Page 13 first paragraph)

Reviewer #2 (Remarks to the Author):

In the present study Chen et al. investigated the role of the surface epitope WWOX286-299 and concluded from their results that this epitope is responsible for repelling WWOX-negative cancer cells. In brief, this is an interesting study. However, some parts of the manuscript are not really

consistent and it seems that as much data as possible were summarized here. Different cell lines were compared with each other making it difficult to conclude that WWOX286-299 is truly responsible for repelling. Likewise, too much previous work has been cited in the results section, which rather belongs to the Introduction or Discussion section. I am very sorry, but the manuscript is not yet suitable for publication in its present form.

Answer: Thank you for your outstanding efforts. As per your request, we have thoroughly revised this manuscript. Please note that the key changes is mentioned in the beginning of this letter. In brief, we have categorized cells into 1) functional WWOX-expressing (WWOXf) cells and 2) WWOX-dysfunctional or -deficient (WWOXd) cells:

i) WWOXf cells are functional in carrying out calcium influx, non-apoptotic bubbling cell death (BCD), and cell migration repellence for WWOXd cells.

ii) WWOXd cells are capable of undergoing explosion in response to stress stimuli, without executing calcium influx. The cells may migrate backward upon facing WWOXf cells, kill WWOXf cells without physical contacts, and utilize the ERK/WWOX/I κ B α signaling for survival.

Major concerns:

1. *A major concern is that only one WWOX negative cell line was used for the experiments. At least one additional WWOX negative cancer cell line should be used to substantiate their findings.*

Answer: Please refer to our statement in the beginning of this letter. As the story goes, we have added new data and developed an advanced concept regarding the nature of WWOXf and WWOXd cells. Functional WWOX protein participates in apoptosis and non-apoptotic bubbling cell death (BCD) under stress conditions (ref. #3,4,34,36,39,40). WWOXd cells express truncated or mutant WWOX protein (mutations mainly in the C-terminal SDR domain) or no WWOX protein expression at all.

Expression of WWOX in any cells does not mean that the expressed protein is functional in performing BCD. The stressed WWOXf cell generates a NO-containing balloon, possessing a connecting small stalk, from the nucleus at room temperature (ref. 39). By time-lapse microscopy we showed WWOXf cells exhibit effective calcium ion influx and BCD in response to UV, UV/cold shock, or other stress stimuli (Figure 1m,o; Supplementary Figure 2a,c). We have first reported BCD from experiments conducted at room temperature or down to 4°C (ref. 39). In response to UV or UV/cold shock, WWOXf L929S cells undergo bubbling first, followed by calcium influx and then death. Ceritinib also induces BCD at 37°C (ref. 33). Hyaluronan induces the ectopic Hyal-2/WWOX/Smad4 signaling for BCD at 37°C (ref. 32).

In contrast, WWOXd cells exhibit reduced activity in calcium influx and undergo explosion, but not BCD, in response to stress stimuli at room temperature. We have discovered this for the first time. Explosion appears to start from the nuclear gas, whereas the nature of this type of cell death remains to be further established.

The aforementioned features correlate nicely with the nature of cell migration. The correlation is based on our key findings from our new studies, as follows:

1) Retrograde migration of WWOXd cells correlates positively with cellular sensitivity to UV/cold shock-induced explosion. UV/cold shock induces BCD in WWOXf cells, but causes explosion in WWOXd cells at room temperature or colder. In this study, **functional WWOXf cells** are TNF-sensitive mouse L929S fibrosarcoma cells, human prostate DU145, human colon cancer HCT116 cells, normal human skin cells, tongue SCC4, 9 and 15, and MEF wild type *Wwox* cells. **WWOX-dysfunctional or -deficient cells** (WWOXd) are human breast cancer MDA-MB-231 and MDA-MB-435s cells, mouse melanoma B16F10 cells, TNF-resistant mouse L929R fibrosarcoma cells, human glioblastoma U87-MG and 13-06-MG cells, human neuroblastoma NB69 cells, human neurofibromatosis NF1 cells, and MEF knockout *Wwox* cells. Please see the new Fig. 1 and new Supplementary Fig. 2 for experimental data in calcium influx, BCD and explosion. 18 new pairs of cell migration are shown in the Supplementary Fig. 3a. WWOX protein expression data is shown in the Supplementary Fig. 3b,c. The new Supplementary Table I also have 48 cell pairs for migration assays.

2) WWOXf cells exhibit calcium ion (Ca^{2+}) influx upon challenge with phorbol myristate acetate (PMA), UV/cold shock, ceritinib, hydrogen peroxide, and many chemotherapeutic chemicals (new Fig. 1m-q; Supplementary Figure 2a-d). In contrast, under similar conditions, the calcium influx function is less efficient in the WWOXd cells (new Fig. 1m-q; Supplementary Figure 2a-d).

3) WWOXd cells induce apoptosis of WWOXf cells from a distance. Similarly, when stably seeded WWOXf L929S cell monolayers are suddenly impacted by WWOXd MDA-MB-231 cells from suspension, L929S cells undergo apoptosis and MDA-MB-231 cells survive (new Figure 2). As impacted by MDA-MB-231 cells from suspension, L929S cells showed a rapid increase in TCF/LEF promoter activation, followed by reduction 15 hr later. In parallel, MDA-MB-231 cells exhibited a time-related slow increase in the promoter activation as impacted by L929S cells from suspension. In addition, survival signaling of I κ B α /ERK/WWOX occurred in MDA-MB-231 cells but not in L929S cells.

Please note the original Table I, there are 27 experiments, using 8 cell lines plus each indicated treatment. Now, Table I has 48 cell-pair experiments. Please refer to Supplementary Fig. 2 for additional cell lines and their responses to UV or indicated stress stimuli, along with calcium influx. Here are examples for L929S versus L929R:

As shown in new experiments in the Figure 1 and Supplementary Fig. 2, WWOXf L929S fibroblasts versus WWOXd L929R fibroblasts. L929S is sensitive to TNF-killing whereas L929R is resistant (Chang, J. Biol. Chem. 270,7765-7772, 1995). L929R was derived from L929S. Please note that L929S was labeled as L929 in our original manuscript. L929R cells migrate reluctantly toward L929S cells, or L929R cells may undergo retrograde migration to run away from L929S cells (Figure 1a-c, e-g, i-k). Occurrence of L929S cell death as induced by L929R cells is shown (Figure 1d,h,l). In addition, as requested by the Reviewer 1, we have conducted the experiments under 0, 2, and 10% fetal bovine serum (FBS) in the RPMI culture medium, respectively. The data showed that the higher the FBS concentration, the less the L929S mobility. The observations suggest that serum factors or adhesion proteins reduce the mobility of L929S. FBS levels do not

affect L929R mobility. Still, L929R cells migrate reluctantly or undergo retrograde migration upon facing L929S (new Figure 1 and new Supplementary Videos 1 to 3).

We reported that L929S cells express functional WWOX and undergo non-apoptotic bubbling cell death (BCD) in response to UV and cold shock (ref. 39). In contrast, L929R cells undergo explosion in response to UV/cold shock. Generation of nitric oxide-containing bubbles probably depends upon the functional and structural integrity of the SDR domain of WWOX. *WWOX* gene encoding the SDR domain tends to undergo mutation and deletion (ref. 4-6).

2. *It remains unclear why the authors have presented data summarized in Fig. 1. Here, data for SCC9, SCC15 cells, MEF cells and HCT116 cells are shown, whereas the MDA-MB-231 breast cancer cell line was used for most of the following experiments. Please explain. In this context it remains ambiguous why SCC9 cancer cells were defined as WWOX negative even though WB data in ref#38 clearly showed that a weak WWOX expression was detected in SCC9 cells. Maybe the authors could also comment on this. Furthermore, it remains unclear why Hyal-2 and WWOX immunoelectron microscopy data of HCT116 were shown here because this has already been done in ref#7. Please explain.*

Answer: Thanks. A great point.

1) Regarding “*..Here, data for SCC9, SCC15 cells, MEF cells and HCT116 cells are shown, whereas the MDA-MB-231 breast cancer cell line was used for most of the following experiments...*”, we understand the awkward arrangement. As mentioned above, we have expanded many more experiments and carefully rearranged many figures.

2) Regarding “*..In this context it remains ambiguous why SCC9 cancer cells were defined as WWOX negative even though WB data in ref#38 clearly showed that a weak WWOX expression was detected in SCC9 cells....*”, we agree your point. Please refer to our explanation shown above and in the beginning regarding WWOXf and WWOXd cells. SCC4, 9 and 15 cells are indeed WWOXf. Please refer to new data in the new Supplementary Fig. 5a-f.

3) Regarding “*.... it remains unclear why Hyal-2 and WWOX immunoelectron microscopy data of HCT116 were shown here because this has already been done in ref#...7*”, we have deleted this figure at your request. In response to the Reviewer #1, we have carried out WWOX and T β RII colocalization in the lipid raft, and determined that both C-terminal SDR domain and D3 tail are needed for the localization of WWOX in the lipid raft (new Fig. 4). By using organs, we showed the complex of WWOX, Hyal-2 and T β RII in the lipid raft (new Fig. 4). Our supporting data showed that each component of the complex is needed for regulating cell migration.

3. *Likewise, line 89-91: “WWOX-deficient human MDA-MB-231 breast cancer cells migrated backward upon facing WWOX-positive mouse L929 fibroblasts (Suppl Video 1)”. Appropriate data have been already shown in Ref#37 (Fig. 5a: MDA-MB-231 underwent retrograde migration upon encountering L929). Please explain.*

Answer: As requested, the original Supplemental Video S1 has been replaced with “MEF knockout *Wwox*^{-/-} cells versus wild type MEF cells” using 10% FBS/medium for time-lapse microscopy. The reason is that Reviewer #1 has requested us to run cell pair migration assays

using different concentrations of FBS to prove that the retrograde migration cannot be abolished by serum factors. The entire experiments are shown in the Fig. 1a-l and Supplementary Fig. 1d-g. Please note that the higher the serum concentration, the slower L929S cells migrate, suggesting that serum factors participate in limiting L929S migration. L929S cells belong to the WWOXf group.

4. Even though it seems to be clear that MDA-MB-231 breast cancer cells lack WWOX expression appropriate Western Blot data should be presented. Likewise, appropriate TbetaRII and Hyal2 WB data should be presented since TGF-beta binds to b1oth.

Answer:

1) **WWOX expression:** As requested, we have added the WWOX protein expression profiles from many cell lines by Western blot (new Supplementary Fig. 3b,c) and their migratory behaviors by pairing WWOXf and WWOXd cells are shown (new Supplementary Fig. 3a). Please note that the WWOX protein levels of MDA-MB-231, MDA-MB-435s and MCF7 breast cancer cells have also been documented (ref. 35). WWOX is inducible upon treating MDA-MB-231 cells with 5-aza-2' deoxycytidine, an inhibitor of DNA methylation (ref. 38). Altogether, there is **no** positive correlation between WWOX protein levels and their control of retrograde migration. That is, functional WWOX is required to control WWOXf cells to respond properly to calcium influx and BCD. WWOXf cells migrate collectively and fend off invading WWOXd cells.

2) **TβRII and Hyal-2 expression:** We have examined many cell lines for their expression for WWOX and TβRII (new Supplementary Fig. 3b,c). Some cells have good expression, whereas certain cells do not. Both WWOX and TβRII are low in the human normal skin fibroblasts. These cells are WWOXf. However, skin neurofibromatosis YMY-NF1 cells from the same patient have increased levels of WWOX and TβRII (new Supplementary Fig. 3c). YMY-NF1 cell is WWOXd.

Because of the difficulty in using cancer cell lines, we decided to use normal organs from mice and showed the presence of the WWOX/Hyal-2/TβRII complex in the lipid raft (new Fig. 4c-e). We will carry out additional experiments to further elucidate the functional relationship between WWOX, Hyal-2 and TβRII in regulating cell-to-cell regulation and migration and will publish later on.

5. The authors should show present appropriate migration data for MDA-MB-231 cells with TbetaRII IgG. Otherwise, it cannot be concluded that MDA-MB-231 cells behave similar to MEF-/- cells.

Answer: As requested, we have carried out the experiments. Data are shown in the new Fig. 3 and Supplementary Fig. 8a-d. Three cell pairs of WWOXf and WWOXd cells were used 1) MEF wild type versus *Wwox* knockout, 2) L929R versus L929S, and 3) MDA-MD-231 versus L929S.

In brief, upon encountering WWOXf cells from a distance, WWOXd cells are able to induce apoptosis of WWOXf cells. This is due to increased redox activity in WWOXf cells (Fig. 5b; see ref. 38). In contrast, by using antibodies against TβRII or pY33-WWOX in pretreating WWOXd cells, WWOXf cells induce apoptosis of WWOXd cells (Fig. 3b-g and Supplementary Fig. 8). Under this condition, WWOXd cells become stationary (new Fig. 3e,g and Supplementary Fig. 8). When antibody was included in the coculture of WWOXd and WWOXf cells during migration,

both cells have enhanced migration and invaded each other's area (Fig. 3d,g and Supplementary Fig. 8). No cell death was observed.

6. *The “Abstract” has to be rewritten since too much previous work is summarized here (e.g., line 22-24: “To compromise, WWOX- cells secrete” and line 26-27: “In contrast, when WWOX+ cells were treated ...). These passages are rather related to the findings of ref#37, but not to the data presented here.*

Answer: As requested, we have thoroughly revised the abstract.

7. *Fig. 4a: The authors should describe more in detail what can be seen on the Western Blot. As written in line 184-185 “stable transfectants of MDA-MB-231 cells expressing EGFP-tagged SDR domain or EGFP only” were established. However, it seems that additional EGFP-tagged proteins were established, such as EGFP-WWOX1-21, EGFP-WWOX7-21 and EGFP-WW domains. Please explain. In this context it remains unknown which housekeeping gene has been used for control purposes.*

Answer: Regarding the constructs for expressing EGFP-WWOX1-21, EGFP-WWOX7-21 and EGFP-WW domains, we did use these expression constructs for establishing stable transfectants and ran migration assays. We did show the data for a stable transfectant of MDA-MB-231-expressing EGFP-WWOX7-21(gre) in the original Table I, exp 26 and 27 (new Table I, exp 27 and 28). The results showed EGFP-WWOX7-21 provided greeting signal and enhanced cell migration. EGFP-WWOX1-21 had similar results as gre, and EGFP-WW domains had no effect (data not shown).

We will make additional stable transfectants of MDA-MB-231 cells for expressing EGFP-tagged WWOXD3, and WWOX Δ D3. In our recent publications (ref. 16,33), we have shown that WWOX7-21 and WWOX7-11 peptides are potent in blocking cancer growth. We will compare our *in vitro* and *in vivo* findings using peptides versus the stable transfectants. Meanwhile, we will elucidate the underlying mechanism (e.g. phosphorylation at specific sites or mutation in epitopes). We will publish our observations later on.

8. *For me it is difficult to conclude that Flotillin is colocalized with SDR domain in lipid rafts because the resolution of the images is too low and the brightness of the green and red fluorescence is too high. In this context, it remains unclear why the green and red intensity of Fig. 4b is much higher than of Suppl. 5a and 5e. Conceivably, this might be attributed to the pdf conversion process.*

Answer: We have determined which region or domain in WWOX that is responsible for localizing in the lipid raft. In the new experiments, we have determined that in addition to the SDR domain, the C-terminal D3 tail has a strong affinity for lipid raft (new Fig. 4b). See the colocalization scores for SDR or D3 with lipid raft (new Fig. 4b). Also, binding of WWOX with flotillin-2 *in vivo* was confirmed by co-immunoprecipitation using organ lysates (new Fig. 4c-e).

Regarding “*In this context, it remains unclear why the green and red intensity of Fig. 4b is much higher than of Suppl. 5a and 5e.*”, please note that the smaller the pixel (or the smaller the

picture) the denser the color intensity. The original Fig. 4b is now Fig. 7b with slight enlargement. The original Supplementary Fig. 5a to 5e is now new Supplementary Fig. 11a to 11e (figures enlarged).

9. *The authors should explain more in detail why MEF cells have been used for data summarized in Fig. 7 and not MDA-MB-231 cells. Likewise, it remains unclear why additional experiments with TGF-beta2 have been done. This should be briefly explained. Likewise, appropriate experiments with TGF-beta2 and MDA-MB-231 cells should be performed. Moreover, it would be great if the authors could briefly discuss why TGF-beta2 suppressed the migration of knockout MEF cells (Fig. 7b, c). This is missing in the discussion.*

Answer: Thank you for raising this question. As we have added too many new data in this revised manuscript, we have decided to delete the original Fig. 7d and g. Regarding your concern, we will thoroughly examine the effect of TGFβs on *Wwox* MEF cells and MDA-MB-231 cells regarding their proliferation and migration. We will publish new findings later on.

As requested, we have added comments on TGFβ/TβRII in the Discussion (page 22, second paragraph). We will check our available gene expression chip data for the MEF wild type and the *Wwox* knockout cells and identify potential candidates. After this, we will treat these MEF cells with TGFβ1 or TGFβ2, and then examine their gene expression profiles by gene chip or NGS.

10. *What is the effect of TGF-beta2 on the proliferation of the MEF wildtype and knockout cells?*

Answer: As mentioned above, we have expanded current study tremendously. Accordingly, we will carry out the effect of TGFβ2 on the proliferation of MEF wildtype and knockout cells in the near future. We will publish the study results then.

11. *Statistical analysis: I strongly recommend to the authors to recalculate the statistics using an appropriate method (either one-way ANOVA (if data are Gaussian distributed) or a nonparametric test (e.g., Kruskal-Wallis) with an appropriate post-hoc test).*

Answer: As requested, all our new data have been analyzed using one-way ANOVA. Please refer to the legend in each figure.

Minor comments:

1) *line 89-91: “WWOX-deficient human MDA-MB-231 breast cancer cells migrated backward upon facing WWOX-positive mouse L929 fibroblasts (Suppl Video 1)”. Appropriate data have been already shown in ref #37 (Fig. 5a: MDA-MB-231 underwent retrograde migration upon encountering L929).*

Answer: As requested, the original video has been deleted. The new Video S1 is “Retrograde migration of MEF knockout *Wwox*^{-/-} cells upon encountering wild type *Wwox*^{+/+} assayed under 10% FBS/medium”. Many wild type cells underwent apoptosis. Please also see Answer #3 shown above.

2) *Error bars are missing in Fig. 1c, Fig. 2d, Fig. 4i, and Fig. 7g. Please correct.*

Answer: As requested, we have fixed the aforementioned standard error bars. They are original 1c to new 3a (with added new data) and Supplementary Fig. 7 (with added new data); Fig. 2d to new 5g; Fig. 4i to new 7i. Please note that the original Figure 7 has been largely deleted, simply because we have expanded the manuscript a lot.

3) *Mat & Method section: cell culture conditions for SCC-9 and SCC-15 cells are missing.*

Answer: As requested, the culture conditions for SCC4, 9, and 15 have now been added in the Methods. Please also refer to new data in Please note a panel of WWOXf and WWOXd cell lines have been added in the study and their culture conditions have been clearly described in the Methods Section.

4) *Line 410: “liver and lung cells were”: no experiments with liver cells were performed here.*

Answer: We have checked the original data for the primary liver cells as well as the cell stocks in the liquid nitrogen. We have decided to delete the statement. The primary liver cells are not in good quality. Our apologies.

5) *Instead of referring to previous published work a brief description for Western Blot experiments and immunofluorescence microscopy including the used antibodies and fluorochromes should be given here.*

Answer: As requested, we have briefly increased the description in the text for Western blot, immunofluorescence microscopy, and the nature of secondary antibodies.

6) *Line 293: “MDA-MB-231 cells to form cancer stem cell-like spheres”. It is commonly known that MDA-MB-231 breast cancer cells rather form cell clumps than spheres (Manuel Iglesias et al. 2013, PLoS ONE 8:e77281).*

Answer: Thank you. The text and data have been deleted. Actually, these cell clumps can be stained positively with a live fluorescent “BioTracker 529 Green Pluripotent Stem Cell Dye” (Millipore). This agent binds mitochondria in iPS/ES cells, but not in differentiated cells. Cells in the clumps are positive with several stem cell markers. Of course, additional stringent criteria are needed to facilitate the definition that these cells are cancer stem cells. In addition, we are currently testing the effect of WWOX microenvironment in affecting the formation of normal and cancer stem cell spheres. We will publish our discovery later on.

Our Additional Changes

1. Discussion section has been thoroughly revised. We have added new discoveries, importance concepts and perspectives, and future directions.
2. Please note that the original L929 is now L929S. This cell line is sensitive to TNF-mediated cell death. L929R is derived from L929S. L929R cells are resistant to TNF killing.

3. Please note that the title of our manuscript has been changed. New title is “Cell surface epitope WWOX286-299 in normal cells is responsible for repelling invading WWOX-negative or -dysfunctional cancer cells“.

We hope that the additions and revisions are satisfactory.

Sincerely,

Nan-Shan Chang, Ph.D.

Distinguished Professor & Ex-Director

Institute of Molecular Medicine, National Cheng Kung University, Taiwan

Adjunct Professor

NYS Institute of Basic Research for Developmental Disabilities, Staten Island, NY

China Medical University, Taichung, Taiwan

Email: changns@mail.ncku.edu.tw; wox1world@gmail.com

Reviewers' comments:

Reviewer #1 (Remarks to the Author):

Recommendation: This research represents an interesting and potentially important series of experiments investigating the roles of different epitopes of the tumor suppressor Wwox gene in cell recognition, and how TGF β is involved in this process. Most of the data support the conclusion. But some specific mechanisms still need further research and there are some minor issues that need correction.

Specific Comments:

- When verifying the cell migration experiment in this manuscript, the serum concentration of the culture medium is 2% to inhibit the proliferation and division of cells. In the process of tumor development, cancer cells proliferate indefinitely. Does the phenomenon of Wwox-positive cells repelling invading Wwox-negative cancer cells still exist?
- The anchoring of Wwox on the membrane can be achieved by combining with Hyal-2 or self-polymerization. How does these two anchoring modes affect Wwox to perform different functions? Please verify, which would increase the clarity and significance their results?
- To determine the effect of TGF β signaling on repelling Wwox-negative cells, Wwox negative and Wwox positive + T β Rii IgG were used for comparison. Is this rigorous? I suggest that Wwox negative, Wwox positive, and Wwox positive + T β Rii IgG should be used to illustrate this conclusion.
- Wwox 7-21 peptide promote the greeting and migration of cells. Meanwhile, the phosphorylation of S14 could inhibit this promotion either. The result shows us that S14 plays an important role in the process of cell greeting and migration. However, the 11-20 peptide has no effect on cell migration. Why?
- In Figure 1h, the peptide was tested for toxicity to cell growth using 2% FBS. While 2% FBS only guarantees the survival of cells and inhibits their division, In this regard it is unclear to me what is the significance of detecting the cell cycle under different peptide treatments under 2% FBS?
- In Figure 2, the coating efficiency of peptides was detected by ELISA. The four peptides (gre, pS14gre, repl, pY287repl) were not completely consistent. Should the influence of the four peptides on cell migration be tested under the same coating efficiency?
- In Figure 3c, the gre signal should be localized in the cell membrane, but appears in the nucleus in the merge image. Please specify
- What effect does the SDR domain have on the migration of the cell itself? From the comparison of Figures 4b and 4c, compared with EGFP, cells expressing SDR basically do not migrate.
- Please explain why there seems to be no statistical differences in Figure s2, s3,1c,2d and 7g.
- In the schematic diagram shown in Figure 8d, phosphorylation at site 33 of Wwox leads to the exposure of the repl peptide, but it seems that there are no same issues in the manuscript to support this conclusion.
- What role does the 287 site of Wwox play in cell migration? On line 183 of the manuscript, phosphorylation of Y287 can reduce cell migration by nearly 50%. Whereas on line 201 of the manuscript, Y287 phosphorylation is not needed for cell repulsion. Why?

Reviewer #2 (Remarks to the Author):

Comments to Manuscript Chen et al.

In the present study Chen et al. investigated the role of the surface epitope WWOX286-299 and concluded from their results that this epitope is responsible for repelling WWOX-negative cancer cells. In brief, this is an interesting study. However, some parts of the manuscript are not really consistent and it seems that as much data as possible were summarized here. Different cell lines were compared

with each other making it difficult to conclude that WWOX286-299 is truly responsible for repelling. Likewise, too much previous work has been cited in the results section, which rather belongs to the Introduction or Discussion section. I am very sorry, but the manuscript is not yet suitable for publication in its present form.

Major concerns:

1) A major concern is that only one WWOX negative cell line was used for the experiments. At least one additional WWOX negative cancer cell line should be used to substantiate their findings.

2) It remains unclear why the authors have presented data summarized in Fig. 1. Here, data for SCC9, SCC15 cells, MEF cells and HCT116 cells are shown, whereas the MDA-MB-231 breast cancer cell line was used for most of the following experiments. Please explain.

In this context it remains ambiguous why SCC9 cancer cells were defined as WWOX negative even though WB data in ref#38 clearly showed that a weak WWOX expression was detected in SCC9 cells. Maybe the authors could also comment on this.

Furthermore, it remains unclear why Hyal-2 and WWOX immunoelectron microscopy data of HCT116 were shown here because this has already been done in ref#7. Please explain.

Likewise, line 89-91: "WWOX-deficient human MDA-MB-231 breast cancer cells migrated backward upon facing WWOX-positive mouse L929 fibroblasts (Suppl Video 1)". Appropriate data have been already shown in Ref #37 (Fig. 5a: MDA-MB-231 underwent retrograde migration upon encountering L929). Please explain.

3) Even though it seems to be clear that MDA-MB-231 breast cancer cells lack WWOX expression appropriate Western Blot data should be presented. Likewise, appropriate TbetaRII and Hyal2 WB data should be presented since TGF-beta binds to both.

4) The authors should show present appropriate migration data for MDA-MB-231 cells with TbetaRII IgG. Otherwise, it cannot be concluded that MDA-MB-231 cells behave similar to MEF^{-/-} cells.

5) The "Abstract" has to be rewritten since too much previous work is summarized here (e.g., line 22-24: "To compromise, WWOX- cells secrete" and line 26-27: "In contrast, when WWOX+ cells were treated ...). These passages are rather related to the findings of ref#37, but not to the data presented here.

6) Fig. 4a: The authors should describe more in detail what can be seen on the Western Blot. As written in line 184-185 "stable transfectants of MDA-MB-231 cells expressing EGFP-tagged SDR domain or EGFP only" were established. However, it seems that additional EGFP-tagged proteins were established, such as EGFP-WWOX1-21, EGFP-WWOX7-21 and EGFP-WW domains. Please explain. In this context it remains unknown which housekeeping gene has been used for control purposes.

7) For me it is difficult to conclude that Flotillin is colocalized with SDR domain in lipid rafts because the resolution of the images is too low and the brightness of the green and red fluorescence is too high. In this context, it remains unclear why the green and red intensity of Fig. 4b is much higher than of Suppl. 5a and 5e. Conceivably, this might be attributed to the pdf conversion process.

8) The authors should explain more in detail why MEF cells have been used for data summarized in Fig. 7 and not MDA-MB-231 cells. Likewise, it remains unclear why additional experiments with TGF-beta2 have been done. This should be briefly explained. Likewise, appropriate experiments with TGF-beta2 and MDA-MB-231 cells should be performed. Moreover, it would be great if the authors could briefly discuss why TGF-beta2 suppressed the migration of knockout MEF cells (Fig. 7b, c). This is missing in the discussion.

9) What is the effect of TGF-beta2 on the proliferation of the MEF wildtype and knockout cells?

10) Statistical analysis: I strongly recommend to the authors to recalculate the statistics using an appropriate method (either one-way ANOVA (if data are Gaussian distributed) or a nonparametric test (e.g., Kruskal-Wallis) with an appropriate post-hoc test).

Minor comments:

1) line 89-91: "WWOX-deficient human MDA-MB-231 breast cancer cells migrated backward upon facing WWOX-positive mouse L929 fibroblasts (Suppl Video 1)". Appropriate data have been already shown in ref #37 (Fig. 5a: MDA-MB-231 underwent retrograde migration upon encountering L929).

2) Error bars are missing in Fig. 1c, Fig. 2d, Fig. 4i, and Fig. 7g. Please correct.

3) Mat & Method section: cell culture conditions for SCC-9 and SCC-15 cells are missing.

4) Line 410: "liver and lung cells were": no experiments with liver cells were performed here.

5) Instead of referring to previous published work a brief description for Western Blot experiments and immunofluorescence microscopy including the used antibodies and fluorochromes should be given here.

6) Line 293: "MDA-MB-231 cells to form cancer stem cell-like spheres". It is commonly known that MDA-MB-231 breast cancer cells rather form cell clumps than spheres (Manuel Iglesias et al. 2013, PLoS ONE 8:e77281).

April 13, 2021

REVIEWERS' COMMENTS:

Reviewer #1 (Remarks to the Author):

The authors have properly addressed all my concerns. It is suitable to be accepted now.

Answer: Thanks again for your outstanding efforts and enthusiasm.

Reviewer #2 (Remarks to the Author):

Comments to revised Manuscript Chen et al.

Due to the corrections the authors have made the overall quality of the manuscript has been significantly improved. I also acknowledge the authors work that they have done additional experiments. However, I have some last comments.

Answer: We greatly appreciate your enthusiasm, outstanding contributions, and great efforts.

1) For me, Western Blot data in Fig. 7a are still not labeled clearly. The green arrow depicts the GFP fusion proteins in the different cell lines, but what is marked with the red asterisk? I presume that this is most likely the house-keeping protein. However, this information is missing in the text/figure legend. Likewise, a red asterisk marked band is missing in lane 6. I also missed a detailed protocol how GFP variants were constructed. It would also be great if the authors could indicate which of the used GFP constructs is “gre” and “repl” since this would be much easier for the readership.

Answer: Regarding “...*The green arrow depicts the GFP fusion proteins in the different cell lines, but what is marked with the red asterisk? I presume that this is most likely the house-keeping protein. However, this information is missing in the text/figure legend. Likewise, a red asterisk marked band is missing in lane 6.*”, please accept our apologies. In the beginning, we got confused with your mentioning of “house-keeping protein”. These proteins, now marked in red stars from left to right, are non-specific proteins bound by the GFP antibody from Santa Cruz Biotechnologies. We are sorry for forgetting to mention about this in the figure legend. Please refer to the revised legend of new Figure 7a (page 36).

Regarding “... *I also missed a detailed protocol how GFP variants were constructed. It would also be great if the authors could indicate which of the used GFP constructs is “gre” and “repl” since this would be much easier for the readership.*” Our apologies again. We did forget. Please refer to a full description in the Methods section regarding how GFP-WWOX cDNA constructs and the WWOX variant constructs were made (page 25).

We have added a simplified schematic graph (from Fig. 5a) to illustrate the regions of gre and repl in the Fig. 7a. We believe that this will provide better reading.

2) *I am sorry, I have overseen this in my first comments. It must be noted that the use of the MDA-MB-435S cell line is still controversial due to the uncertainty as to whether it is a breast cancer cell line or a melanoma cell line. To avoid potential conflicts, MDA-MB-435S cells should be rather named as “cancer cells” and not as “breast cancer cells”. Otherwise, the authors should include a short statement in the discussion section briefly discussing the MDA-MB-435S mammary/ melanoma problem.*

Answer: Thank you so much for mentioning this to us. As requested, we have removed the “breast cancer” for MDA-MB-435S and now use “cancer cell” only (pages 5, 7, 24), and have added a short statement in the Methods describing the origin of MDA-MB-435S (page 24). No changes are needed in the Supplementary Materials.

Additional changes

Figure 4e: We have replaced the last data and added the bands from heavy chains and Hyal-2 bands in this figure.

Full-length Western gels: Please see page 43 to 45 of the Supplementary Materials regarding the full-length Western blots.

Supplementary Figure 1b: An additional new data is added as a replicate (page 2, Supplementary Materials). The original data was carried out 10 years ago. My student has graduated and left our laboratory then. She could not find the original data spreadsheet due to her very old computer. We re-did the work and showed the reproducibility. A new spreadsheet has been uploaded to the system using the new data. Please note that this is the only data we cannot recover for all the needed spreadsheets for this article covering the last 10-year research.

Supplementary Figures 2 and 5: Twenty seven original videos, labeled as Video S1 for Suppl. Fig. 2a, 2b, etc., are attached in the Supplementary Materials. Three videos for “Video S1 for Suppl. Fig. 5” are added. These allow readers to analyze our data.

Supplementary Figure 3: The full-length Western blot data for Suppl. Fig. 3b4 have been added. The original Suppl. Fig. 7, using primary normal human skin fibroblasts and neurofibromatosis NF1 cells, has been deleted. We are currently working on these two cell lines at an advanced stage and will publish new findings later on this year.

Figure style: As per Journal requests, we have changed all the bar graphs according to the Journal guidelines. If space is sufficient, we have added the p -values in each bar graph for the statistics. If not, we have provided additional graphs in the Supplementary Materials under the “Expanded Figures with detailed statistics”, page 45 to 47).

We hope that the revisions are satisfactory.

Sincerely,

Nan-Shan Chang, Ph.D.

Distinguished Professor & Ex-Director

Institute of Molecular Medicine, National Cheng Kung University, Taiwan

Adjunct Professor

NYS Institute of Basic Research for Developmental Disabilities, Staten Island, NY

China Medical University, Taichung, Taiwan

Email: changns@mail.ncku.edu.tw; nschang13827@gmail.com